# Adaptive Probe-based Steering for Robust LLM Jailbreaking

Junxi Chen [1]   Junhao Dong [2✉]   Xiaohua Xie [1✉]

## Abstract

Recent work has demonstrated the potential of contrastive steering for jailbreaking Large Language Models (LLMs). However, existing methods rely on limited and inherently biased contrastive prompts and require laborious manual tuning of steering strength, limiting their robustness and effectiveness. In this paper, we leverage the idea of model extraction to guide the learned steering vectors to approximate the ideal one and propose tuning the steering strength adaptively based on contrastive activations' statistics. Experiments demonstrate that our method notably improves the effectiveness and robustness of probe-based steering, without any extra contrastive prompts or laborious manual tuning. Being an attack paper, this paper focuses on revealing the breakdown of fortified LLMs, raising the average harmfulness score from 6% to 70%. Our code is available at https://github.com/fhdnskfbeuv/adaptiveSteering.

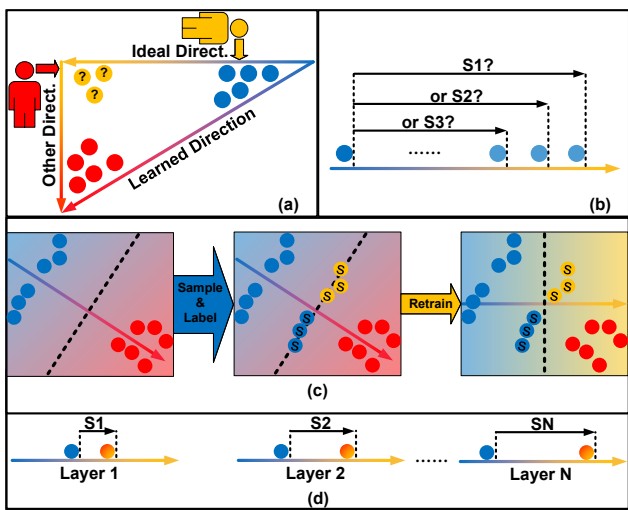

Figure 1. (a) The same learned direction, trained on the same contrastive prompts, can form different concept classifiers, which reveals the existence of multiple coupled directions. (b) Determining the steering strength requires laborious continuous parameter tuning. (c) We propose utilizing adaptive retraining to refine the direction, and (d) leveraging the statistics of contrastive activations to determine the steering strength of each layer.

## 1. Introduction

Large Language Models (LLMs) often undergo alignment (Bai et al., 2022; Rafailov et al., 2023) to ensure their harmlessness. However, research (Zou et al., 2023b; Liu et al., 2024; Carlini et al., 2023; Zou et al., 2023a; Xu et al., 2024; Arditi et al., 2024; Dong et al., 2025b; 2026) has shown that such alignment can be circumvented by certain adversarial techniques, known as jailbreaking. To counter jailbreaking, the community has developed various defense methods (Zou et al., 2024; Qi et al., 2025) that reportedly achieve strong adversarial robustness, often claiming near-zero harmfulness scores.

The key challenge in (truly) improving adversarial robust-

ness (Dong et al., 2024; 2025a) is developing strong attacks (Athalye et al., 2018) that can reveal the **worst-case** robustness. When attacking classification models, given differentiable and goal-correlated proxies (e.g., Cross-entropy Loss) that are readily available and the well-established gradient-based optimization, the community quickly developed strong automated attacks (Madry et al., 2018; Athalye et al., 2018; Croce & Hein, 2020), supporting effective and efficient adversarial robustness evaluation.

However, when the adversarial goal is to induce harmful responses from LLMs, developing strong attacks becomes hard. First, the input space of LLM is discrete, making it hard to apply strong gradient-based optimization. Second, a good proxy is hard to find. So far, at the response level, the community has proposed two proxies: **harmful prefix** (Carlini et al., 2023; Zou et al., 2023b) and **jailbreaking judge** (Chao et al., 2025; Mehrotra et al., 2024). While both have been proven effective against certain LLMs, the former is heuristic and may not be well correlated with LLM's harmfulness (Zhou et al., 2025; Liao & Sun, 2024; Zhu et al., 2024), and the latter is usually non-differentiable,

[1]School of Computer Science and Engineering, Sun Yat-Sen University, China [2]Nanyang Technological University, Singapore. Correspondence to: Xiaohua Xie <xiexiaoh6@mail.sysu.edu.cn>, Junhao Dong <junhao003@ntu.edu.sg>.

*Proceedings of the 43rd International Conference on Machine Learning*, Seoul, South Korea. PMLR 306, 2026. Copyright 2026 by the author(s).

making optimization hard to conduct. These limitations hinder the development of strong automated attacks, even against white-box LLMs.

One promising technique that can mitigate the above-mentioned limitations is contrastive steering[1] (Zou et al., 2023a; Arditi et al., 2024). The contrastive steering is based on the assumption that the concept is represented along a linear direction in the representation space. In practice, such direction is acquired by modeling the activation of LLMs facing paired contrastive prompts and can control white-box LLM's behavior by steering activations along itself. Normally, contrastive steering only requires contrastive prompts **without a target response** and does not require backpropagation, but only forward. Such a target-and-gradient-free characteristic gives contrastive steering a lower barrier than gradient-based prompt attacks (Zou et al., 2023b) and fine-tuning attacks (Qi et al., 2024).

In this paper, we enhance the effectiveness and robustness of probe-based contrastive steering (Xu et al., 2024; Hedström et al., 2025), a concise and controllable instantiation. Our idea is also illustrated in Figure 1. All contrastive steerings consist of two stages: direction searching and strength tuning. For direction searching, we argue that the learned directions contain errors inherently caused by the contrastive prompts. Inspired by model extraction (Tramèr et al., 2016), we propose an algorithm that iteratively samples and annotates steered activations to approximate the ideal directions, **without the need for extra contrastive prompts**. Regarding strength tuning, we find that the commonly used accuracy-based layer selection (Xu et al., 2024) is unstable, and that setting layer-uniform logit targets (Xu et al., 2024; Hedström et al., 2025) overlooks the differences in activation magnitudes across layers, thus leading to oversteering. We propose deprecating accuracy-based layer selection and using contrastive activations' statistics to guide logit target setting. Our contributions are as follows:

1. We propose an iterative direction-refinement algorithm that approximates the ideal steering vector. Such an algorithm requires no extra contrastive prompts and leverages only an extra off-the-shelf annotator.

2. We identify the instability of the widely adopted accuracy-based layer selection and the risk induced by layer-wise uniform logit targets. To address these issues, we deprecate accuracy-based layer selection and utilize the statistics of contrastive activations to guide logit target setting, which improves effectiveness and eliminates the need for laborious continuous-parameter tuning.

3. We evaluate our method on LLMs specifically hard-

ened against jailbreaking[2]. Results demonstrate that our method can elevate the harmfulness scores from 6% to at least 50%, and mostly to more than 70%.

## 2. Preliminary

### 2.1. Problem Statement

The ultimate goal of studying jailbreaking is to build up reliable LLMs. Formally, given a toxicity-detection oracle isToxic($\cdot$), the goal is to solve

$$\min_{\theta} \max_{\Delta\theta} \text{isToxic}(m_{\theta}(\Delta\theta)), \qquad (1)$$

where $\theta$ is the LLM's weight, $\Delta\theta$ is any modification that can be applied to the LLM, and $m_{\theta}(\Delta\theta)$ represents the LLM's state (including but not limited to output texts, hidden states, attention maps, etc.). Here, we refer to the modification as $\Delta\theta$ because both modifications in the input space and the embedding space can be seen as altering $\theta$.

The challenge in solving Equation (1) lies in the inner maximization. On one hand, during the training phase, a weak maximization will lead to sub-optimal robustness (Madry et al., 2018). On the other hand, a weak maximization during the evaluation will lead to a false sense of robustness (Athalye et al., 2018). Thus, this paper aims to better solve the inner maximization.

### 2.2. Autoregressive Decoder-only Transformers

Modern LLMs are mostly autoregressive decoder-only transformers (Vaswani et al., 2017). Vertically, the LLM is composed of stacked decoders with residual connections. Given a hidden state $\mathbf{x}^{(l-1)} \in \mathbb{R}^d$, the $l$-th decoder $\text{Dec}_l(\cdot)$ conducts

$$\mathbf{x}^{(l)} := \mathbf{x}^{(l-1)} + \text{Dec}_l(\mathbf{x}^{(l-1)}). \qquad (2)$$

Horizontally, at each autoregressive step, a $L$-layer LLM maps the input embedding at the $i$-th token position $\mathbf{x}_i^{(0)} \in \mathbb{R}^d$ to the last-layer hidden state $\mathbf{x}_i^{(L)}$. $\mathbf{x}_i^{(L)}$ will be later mapped back to an input embedding $\mathbf{x}_{i+1}^{(0)}$ for the next autoregressive step.

### 2.3. Contrastive Steering

The contrastive steering is based on the assumption that LLMs' behaviors (or personas) can be controlled by steering the hidden states along a linear direction. Given a $L$-layer LLM with hidden size $d$, the contrastive steering conducts

$$\mathbf{x}^{(l)} := \mathbf{x}^{(l)} + \lambda^{(l)} * \mathbf{v}^{(l)}, \qquad (3)$$

where $\lambda^{(l)} \in \mathbb{R}$ controls the steering strength at layer $l$ according to $\mathbf{x}^{(l)}$, and $\mathbf{v}^{(l)} \in \mathbb{R}^d$ is the steering vector.

---

[1]We formally introduce existing methods in Section 3.2.

[2]We leave results on general-purpose LLMs in Appendix C.

As shown in Equation (3), the challenges of designing the contrastive steering lies in **(1)** how to find steering vectors $V = (\mathbf{v}^{(1)}, \ldots, \mathbf{v}^{(L)}) \in \mathbb{R}^{L \times d}$ and **(2)** how to determine the steering strength $\lambda^{(l)}$?

### 2.4. Threat Model

This paper focuses on the worst-case robustness of LLMs against jailbreaking. That is, we only discuss bare LLMs rather than LLM-centered systems (e.g., commercial chatbot) and explore how high the harmfulness score can be under certain constraints.

**Data.**  In terms of the constraint on data, we assume the attacker only has a malicious query set $\mathcal{P}_m$ and a benign query set $\mathcal{P}_b$, *without any responses*[3]. Such data restriction, particularly for malicious queries, is sound, as limited attackers usually try to acquire knowledge from the LLM rather than teach it.

**Model.**  We assume the attacker can access and manipulate all LLM blocks' (e.g., MLP, Self-attention) output during the inference. The attacker can not compute gradients of any LLM's embedding or output value with respect to the LLM's inputs or parameters.

## 3. Method

In this section, we first formally introduce the assumption on which the contrastive steering is based in Section 3.1. Then, we review how existing methods tackle these two challenges and analyze their limitations in Section 3.2. Lastly, we introduce our method in Section 3.3.

### 3.1. Linear Control Assumption

The contrastive steering is based on the assumption that the LLMs' behavior can be controlled by adding vectors to the LLMs' hidden state, which is a simple linear operation. Below, to facilitate our discussion of existing methods and our own, we formally describe the Linear Control Assumption (LCA).

**Definition 3.1** ($\beta$-Indicator). $\beta$-Indicator $\mathbb{I}_\beta(\cdot) : X \mapsto \{0, 1\}$ is an indicator function judging whether the input satisfies a specific behavior (or persona) $\beta$, where $X$ is an arbitrary set, and $\mathbb{I}_\beta(x) = \begin{cases} 1, & \text{if } x \text{ satisfies } \beta, \\ 0, & \text{otherwise.} \end{cases}$

**Definition 3.2** $((V, H, S)$-Similar). Given a vector sequence $V = (\mathbf{v}^{(1)}, \ldots, \mathbf{v}^{(L)})$, a real number sequence $S = (s_1, \ldots, s_L)$, and a similarity metric $\text{score}(\cdot, \cdot) \colon \mathbb{R}^d \times \mathbb{R}^d \mapsto$

---

[3]A line of optimization-based steering (Cao et al., 2024; Dunefsky & Cohan, 2025), which should be viewed as a form of parameter-efficient fine-tuning (PEFT), does not fulfill such a data constraint because they require target responses.

$\mathbb{R}$, a $L$-layer LLM's state $m_\theta(\Delta\theta)$ is $(V, H, S)$-Similar iff. a subset $H$ of its hidden states satisfies $\forall \mathbf{x}_i^{(l)} \in H, \text{score}(\mathbf{x}_i^{(l)}, \mathbf{v}^{(l)}) = s_l$, and is at least $(V, H, S)$-Similar iff. $\forall \mathbf{x}_i^{(l)} \in H, \text{score}(\mathbf{x}_i^{(l)}, \mathbf{v}^{(l)}) \geq s_l$.

Definition 3.2 quantifies the LLM's state. The definition of $\text{score}(\mathbf{u}, \mathbf{v})$ depends on the linear model. For example, $\text{score}(\mathbf{u}, \mathbf{v}) = \mathbf{u} \cdot \mathbf{v}$ for bare vector sequences, and $\text{score}(\mathbf{u}, \mathbf{v}) = \mathbf{u} \cdot \mathbf{v} + b$ if a linear probe with bias $b$ is applied.

**Definition 3.3.**  Given two real number sequences with identical length $A = (a_1, \ldots, a_L)$ and $B = (b_1, \ldots, b_L)$, we say $A > B$, iff. $\forall i, a_i > b_i$, and $A = B$, iff. $\forall i, a_i = b_i$.

Definition 3.3 is introduced to facilitate the comparison of different LLMs' states.

**Assumption 3.4.**  Given a behavior $\beta$, there exists an ideal vector sequence $V^* = (\mathbf{v}^{(1)}, \ldots, \mathbf{v}^{(L)})$ such that, for any arbitrary LLM state $m_\theta(\Delta\theta)$, if $m_\theta(\Delta\theta)$ is $(V, H, S)$-Similar, then $\mathbb{I}_\beta(m_\theta(\Delta\theta))$ is a monotonically increasing function of $S$ on the interval $[\underline{S}, \bar{S}]$.

*Remark* 3.5 (on Assumption 3.4). If Assumption 3.4 holds, the behavioral control problem of LLMs (i.e., maximizing $\mathbb{I}_\beta(m_\theta(\Delta\theta))$) can be reduced to the problem of increasing the similarity between $H$ and $V$. The former problem is difficult to solve due to the complex $\mathbb{I}_\beta(m_\theta(\Delta\theta))$, whereas the latter is simple and often admits a closed-form solution, thereby reducing the difficulty of LLM behavioral control.

### 3.2. Existing Methods

We review existing contrastive steering by elucidating the connection between Assumption 3.4 and their design rationale for **direction searching** and **strength tuning**.

#### 3.2.1. DIRECTION SEARCHING

To search $V^*$, existing methods include two consecutive stages: extracting contrastive hidden states and modeling the difference between contrastive hidden states.

During the extraction stage, given a $L$-layer LLM with hidden size $d$, contrastive steering inputs benign instruction set $\mathcal{P}_b$ and malicious instruction set $\mathcal{P}_m$ to capture the LLM's faithful hidden states $H_b^P = \left\{ \mathbf{a}_i^{(l)} \mid i \in P, l \in \{1, \ldots, L\} \right\} \subset \mathbb{R}^d$ and faithless hidden states $H_m^P = \left\{ \mathbf{b}_i^{(l)} \mid i \in P, l \in \{1, \ldots, L\} \right\} \subset \mathbb{R}^d$ at token positions $P$, respectively. During the modeling stage, a linear model (including Linear Probes (Xu et al., 2024), PCA (Zou et al., 2023a), Mean-in-Differences (Arditi et al., 2024), etc.) is applied to capture the differences between $H_b^P$ and $H_m^P$. The outcome of the modeling stage is a vector sequence $V = (\mathbf{v}^{(1)}, \ldots, \mathbf{v}^{(L)})$.

**Algorithm 1** Iterative Training Set Augmentation with Contrastive Activations

---

**Require:** Contrastive prompts $\mathcal{P}_m$ and $\mathcal{P}_b$, LLM $m(\cdot)$ with $L$ layers, Annotator $\mathbb{I}_{faithful}(\cdot)$ returning $(data, label)$, Total iterations $T$, Token Position $P$, Steering strength $S = \{s^{(l)} \mid l \in \{1, 2, \ldots, L\}\}$

**Ensure:** Final trained probe sets $F_T$

1: $\mathcal{A}_0 \leftarrow m(\{\mathcal{P}_m, \mathcal{P}_b, P\})$
2: Train initial probe sets $F_0$ using $\mathcal{A}_0$
3: **for** $i = 0$ to $T - 1$ **do**
4: $\quad \mathcal{A}_{aug} \leftarrow \mathbb{I}_{faithful}(m(\{\mathcal{P}_m, F_i, P\}))$
5: $\quad \mathcal{A}_{i+1} \leftarrow \mathcal{A}_i \cup \mathcal{A}_{aug}$
6: $\quad$ Train $F_{i+1}$ using $\mathcal{A}_{i+1}$
7: **end for**
8: return $F_T$

---

### 3.2.2. STRENGTH TUNING

After acquiring the directions, the contrastive steering needs to determine the steering strength for each layer. Using the notation from Assumption 3.4, we can say that such strength tuning is about increasing $S$ while determining $\bar{S}$ to bound $S$. $\bar{S}$ is important because large strength tends to induce incoherent responses. Existing strength tuning can be categorized into constant, ablation, and probes.

**Constant.** Constant (Zou et al., 2023a) sets $\lambda^{(l)}$ to a fixed value. Constant does not guarantee that $m_\theta(\Delta\theta)$ will be or be at least $(V, H, S)$-Similar after steering, but trivially increases $\text{score}(\mathbf{x}_i^{(l)}, \mathbf{v}^{(l)})$ during each steering. $\bar{S}$ is ignored.

**Ablation.** Ablation (Arditi et al., 2024; Vu & Nguyen, 2025) sets $\lambda^{(l)} = -\frac{\mathbf{x}_i^{(l)} \cdot \mathbf{v}^{(l)}}{\|\mathbf{v}^{(l)}\|_2^2}$, which will zero out $\mathbf{v}^{(l)}$ in $\mathbf{x}_i^{(l)}$. Ablation guarantees that $m_\theta(\Delta\theta)$ will be $(V, H, \mathbf{0})$-Similar, where $\mathbf{0} \in \mathbb{R}^L$, and assumes $\mathbb{I}_\beta(m_\theta(\Delta\theta)) = 1$ if $m_\theta(\Delta\theta)$ is $(V, H, \mathbf{0})$-Similar.

**Probes.** When linear probes (Xu et al., 2024; Hedström et al., 2025) $f(\mathbf{x})^{(l)} = \mathbf{w}^{(l)} \cdot \mathbf{x} + b^{(l)}$ are used for modeling, one can guarantee that $m_\theta(\Delta\theta)$ is at least $(V, H, S)$-Similar by setting $\mathbf{v}^{(l)} = \mathbf{w}^{(l)}$ and $\lambda^{(l)} = \mathbb{I}(s^{(l)} > b^{(l)} + \mathbf{x}_i^{(l)} \cdot \mathbf{w}^{(l)}) * \frac{s^{(l)} - \mathbf{w}^{(l)} \cdot \mathbf{x}_i^{(l)} - b^{(l)}}{\|\mathbf{w}^{(l)}\|_2^2}$. Since linear probes are assumed to well separate $H_b^P$ and $H_m^P$, it is expected to have $\mathbb{I}(f^{(l)}(\mathbf{x}) \geq 0) \equiv \mathbb{I}_\beta(\mathbf{x})$. $s^{(l)}$ is usually set to a large number to further increase the similarity between $V$ and $H$.

### 3.3. Our Method

As previously mentioned, the contrastive steering primarily consists of direction searching and strength tuning, and thus our improvements are mainly focused on these two parts.

More specifically, we primarily enhance the probe-based steering due to its simplicity and controllability.

### 3.3.1. REFINING BIASED LINEAR PROBE WITH MODEL EXTRACTION

Supposing that there exists a set of linear probes $f^{*(l)}(\mathbf{x}) = \mathbf{w}^{*(l)} \cdot \mathbf{x} + b^{*(l)}$ that is ideal for steering, we can view the direction searching as model extraction (Tramèr et al., 2016) because the goal of direction searching is to obtain a set of linear probes $f^{(l)}(\mathbf{x})$ satisfying $f^{(l)}(\mathbf{x}) \approx f^{*(l)}(\mathbf{x})$, identical to model extraction. Although $f^{*(l)}(\mathbf{x})$ is inaccessible, since it is expected that $\mathbb{I}(f^{*(l)}(\mathbf{x}) \geq 0) \equiv \mathbb{I}_\beta(\mathbf{x})$, we can use $\mathbb{I}_{faithful}(\mathbf{x})$ as a proxy, which can be any reliable jailbreaking judge (Souly et al., 2024) in practice.

From this perspective, contrastive steering is inherently a retraining-based model extraction (Tramèr et al., 2016): annotate activations with the judge, and train a linear model with annotated activations. Such a perspective inspires us to refine the linear model by including more annotated activations. One simple approach is to use more contrastive prompts. However, **specifically for the jailbreaking task we discuss**, since we can hardly get harmful prompts that the aligned LLM can follow, the contrastive prompts must be formed by harmful and harmless prompts, which contain other coupled concept directions (e.g., ethical-to-unethical, or even cake-to-bomb) other than the desired faithful-to-faithless direction. Thus, the extracted model is inherently $\mathbb{I}_{faithful}(\mathbf{x}) * \mathbb{I}_{Noise}(\mathbf{x})$. Here, we argue that $\mathbb{I}_{Noise}(\mathbf{x})$ is practically unignorable because one can train the desired faithfulness judge as well as an ethic (or cake-and-bomb) judge **with the same harmful-harmless contrastive prompts.**

To reduce the noise induced by contrastive prompts, we propose to iteratively augment the training set with more contrastive activations that are annotated by reliable $\mathbb{I}_{faithful}(\mathbf{x})$ and correspond to **same prompts**. Specifically, after we obtain the $i$-th probe sets $F_i = \{f_i^{(l)}(\mathbf{x}) \mid l \in \{1, 2, \ldots, L\}\}$, we input harmful prompts[4], generate responses, and collect activations with the LLM steered by $F_i$. Then, we annotate each activation by judging its corresponding responses with $\mathbb{I}_{faithful}(\mathbf{x})$, and add these annotated activations to the training set for the next iteration. The pseudo code is presented in Algorithm 1.

The proposed iterative algorithm has two hyper-parameters: The token position $P$ and the steering strength $S = (s_1, \ldots, s_L)$. In the main paper, the token position $P$ only contains the position immediately preceding the first response token, aligned with prior works (Zou et al., 2023a; Xu et al., 2024; Vu & Nguyen, 2025) for fair comparison.

---

[4]We explain why we ignore benign prompts in Appendix E.

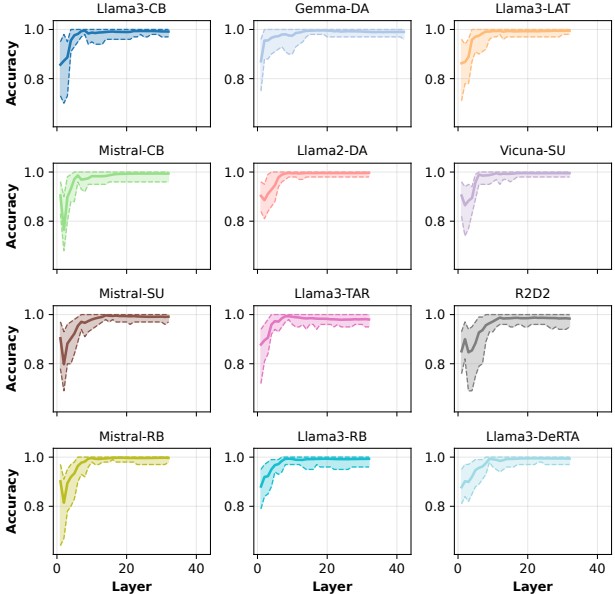

*Figure 2.* Linear probes' accuracy on validation activations across layers. The line represents the mean accuracy, and the shaded area indicates the max and min values of accuracy over 50 random samplings.

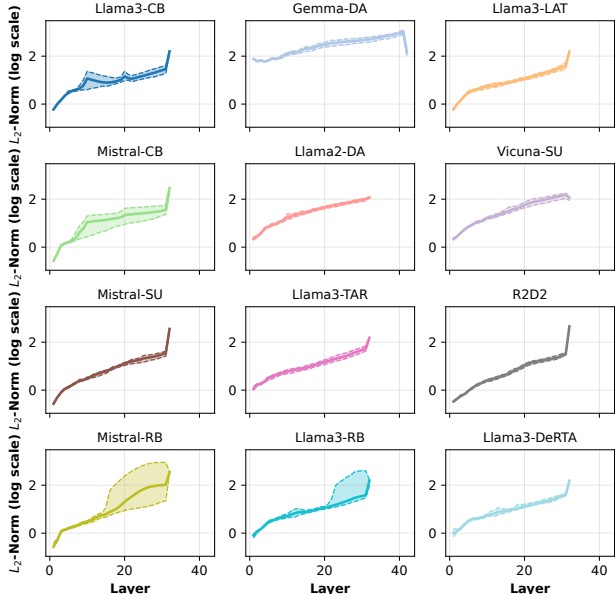

*Figure 3.* $L_2$-norm of activation across layers. The line represents the mean norm, and the shaded area indicates the max and min values of the norm for 100 different activations.

As for the steering strength, we set $s^{(l)} = 0$, which steers activations to just cross the decision boundary. The reason is that augmenting the training set with the points that the current classifier is least certain about has been proven to be more effective and efficient than with randomly sampled points, which is known as **adaptive retraining** (Tramèr et al., 2016) or active learning (Cohn et al., 1994).

### 3.3.2. SIMPLIFYING STRENGTH TUNING

Strength tuning can be a laborious hyper-parameter tuning process since, given an $L$-layer LLM, there are $L$ continuous parameters to be determined. Existing probe-based steering (Xu et al., 2024; Hedström et al., 2025) sets layer-uniform $s^{(l)}$ significantly larger than 0 to ensure altered behavior. Xu et al. (2024) assumes that $f^{(l)}(\mathbf{x})$ having low test accuracy suggests that LCA does not hold at layer $l$. Consequently, they further only steer layers that have test accuracy above a certain threshold (e.g., 90%).

However, we point out that such accuracy-based selection is not robust. We randomly sample 100 paired examples from the contrastive prompts provided by Arditi et al. (2024), split them equally into training and test sets (50% each), and train linear probes. This procedure is repeated 50 times. As shown in Figure 2, the test accuracy is unstable, especially for early layers. Since whether LCA is held at a certain layer is deterministic and objective, we believe that such an accuracy-based perspective is unreliable. Instead, we argue that the exact layerwise discrepancy that should be

considered is the activations' magnitude. We sample 100 prompts and calculate the $L_2$-norm of activations per layer. In Figure 3, we can find that the activation's magnitude increases by orders of magnitude from early layers to late layers. For probe-based steering, when we steer $\mathbf{x}_i^{(l)}$ and set steering strength to $s^{(l)}$, the ratio of the steering vector's norm to $\|\mathbf{x}_i^{(l)}\|_2$ is

$$\frac{s^{(l)} - \mathbf{w}^{(l)} \cdot \mathbf{x}_i^{(l)} - b^{(l)}}{\|\mathbf{w}^{(l)}\|_2 * \|\mathbf{x}_i^{(l)}\|_2} . \tag{4}$$

Clearly, with a layer-uniform $s^{(l)}$, oversteering occurs when $\|\mathbf{x}_i^{(l)}\|_2$ is too small.

Given the limitations described above, we propose to set $s^{(l)}$ according to $\|\mathbf{x}_i^{(l)}\|_2$. A simple and adaptive approach is to utilize the logits of activation $\mathbf{y}_i^{(l)}$ corresponding to the desired behavior. Intuitively, if the probe is ideal, setting the target as the logits of $\mathbf{y}_i^{(l)}$ will guarantee the steered LLM exhibiting the same behavior as the LLM corresponding to $\mathbf{y}_i^{(l)}$. In terms of the activation's magnitude we concern, when we set $s^{(l)} = \mathbf{w}^{(l)} \cdot \mathbf{y}_i^{(l)} + b^{(l)}$, the ratio of the steering vector's norm to $\|\mathbf{x}_i^{(l)}\|_2$ becomes

$$\frac{\|\mathbf{y}_i^{(l)}\|_2 * \cos \theta_{wy} - \|\mathbf{x}_i^{(l)}\|_2 * \cos \theta_{wx}}{\|\mathbf{x}_i^{(l)}\|_2} . \tag{5}$$

Furthermore, since the norm of same-layer activations are of similar order, assuming $\|\mathbf{y}_i^{(l)}\|_2 \approx \|\mathbf{x}_i^{(l)}\|_2$, we can reduce

Equation (5) to $\cos\theta_{wy} - \cos\theta_{wx}$, a value independent of activations' magnitude but depended on the direction.

### 3.3.3. OTHER IMPLEMENTATION DETAILS

**Discarding the Last-layer Activation.** Although also named hidden states in most implementations, the last decoder's activation is different from that of the others. First, steering the last decoder's activation is equivalent to setting logit bias, which can easily induce repetition. Second, the influence of the last decoder's activation is local rather than contextual: It isn't involved in the latter autoregressive steps' calculation but only influences the current steps' unembedding. Given these reasons, we choose to ignore the last decoder's activation, which is a common setting proactively adopted by Zou et al. (2023a), Vu & Nguyen (2025), Arditi et al. (2024), and by Xu et al. (2024) through bug[5].

**Steering All Token Positions.** Since the activations used for direction searching are usually from response token position, some prior works (Xu et al., 2024; Chen et al., 2025) steer only response tokens. Yet, if we view the steering vectors as part of LLM's architecture and weights, it is trivial that we should steer activations at all token positions. In particular, probe-based steering can be seen as a rank-1 LoRA (Hu et al., 2022) with fixed bias:

$$\mathbf{x}_i^{(l)} := \underbrace{\mathbf{I}_d\mathbf{x}_i^{(l)}}_{W_0 x} + \underbrace{(-\hat{\mathbf{w}}^{(l)}\hat{\mathbf{w}}^{(l)\mathrm{T}}\mathbf{x}_i^{(l)})}_{BAx} + \underbrace{\frac{s^{(l)} - b^{(l)}}{\|\mathbf{w}^{(l)}\|_2^2} * \mathbf{w}^{(l)}}_{\text{Fixed Bias}}$$

(6)

and can also be seen as a MLP-based adapter:

$$\mathbf{x}_i^{(l)} := \mathbf{x}_i^{(l)} + \underbrace{\hat{\mathbf{w}}^{(l)}}_{\text{up\_proj}}(\mathrm{ReLU}(\underbrace{-\hat{\mathbf{w}}^{(l)\mathrm{T}}\mathbf{x}_i^{(l)}}_{\text{down\_proj}} + \underbrace{\frac{s^{(l)} - b^{(l)}}{\|\mathbf{w}^{(l)}\|_2}}_{\text{Bias}})),$$

(7)

where $\hat{\mathbf{w}}^{(l)} = \frac{\mathbf{w}^{(l)}}{\|\mathbf{w}^{(l)}\|_2}$. Thus, we argue that we should treat the steering vector as a normal parameter efficient adapter, and thus, applying all-position steering is crucial.

## 4. Experiment

### 4.1. Setups

**Datasets.** We use the first 100 pairs of contrastive prompts provided by Arditi et al. (2024) for direction searching. We use the first 100 harmful prompts of StrongReject (Souly et al., 2024) and of Harmbench (Mazeika et al., 2024) labeled "Standard", resulting in 200 harmful prompts for evaluation.

---

[5]Xu et al. (2024) mistakenly takes activations from layer $l$ as layer $l + 1$ and consequently discards the last decoder's. The code we adopt in this paper is corrected.

**Models.** We primarily focus on LLMs that are specifically designed to defend jailbreaking. We include Circuit Breaker (CB) (Zou et al., 2024), Deep Alignment (DA) (Qi et al., 2025), RepBend (RB) (Yousefpour et al., 2025), R2D2 (Mazeika et al., 2024), SafeUnlearning (SU) (Zhang et al., 2024), TAR (Tamirisa et al., 2025), DeRTA (Yuan et al., 2025) and Latent Adversarial Training (LAT) (Sheshadri et al., 2025), resulting in 12 fortified LLMs in total. All LLMs use greedy decoding throughout our experiment.

**Metrics.** All judges we use are LLM-based, including Fine-tuned LLM (SRF) provided by Souly et al. (2024), a rubric judge (SR) powered by StrongReject's prompt (Souly et al., 2024) and Qwen-Plus (Yang et al., 2025), and the HarmBench classifier (HB) (Mazeika et al., 2024). **We employ these three judges because their design and training emphasize the helpfulness of responses, rather than focusing solely on refusal.** All judges' output scores range from 0 to 1. The maximum length of responses during evaluation is 512. **Since we use SRF for direction searching, we believe using more judges than SRF is crucial for mitigating potential biases (an effect similar to reward hacking) in evaluation.**

**Baselines.** The baselines that we compare all belong to contrastive steering. We include RepE (Zou et al., 2023a), Refusal Direction (RD) (Arditi et al., 2024) that has both constant-based (RD-C) and ablation-based (RD-A) strength tuning, SCAV (Xu et al., 2024), and Angular Steering (Vu & Nguyen, 2025). **All baselines use the same data as ours.** Settings are in Table 10.

**Hyper-Parameters.** During the direction searching, we set $s^{(l)} = 0$ and $T = 20$. The token position we choose is the position immediately preceding the first response token, a common setting adopted by most baselines (Zou et al., 2023a; Xu et al., 2024; Vu & Nguyen, 2025). We split the 100 pairs of contrastive prompts into a training set and a validation set in a 50:50 ratio. 50 pairs are used for Algorithm 1, and the other 50 harmful prompts are used for validation to pick the best direction. The remaining 50 benign prompts are of no use. The maximum response length during the direction searching (including validation) is 256. We chose SRF as the annotator, where samples scored less than 0.05 are classified as negative and samples scored larger than 0.6 are classified as positive. The rest is discarded due to the relatively noisy scoring. Class balance is applied for probe training. During the inference (including validation), for each layer, we choose the largest known logit of contrastive activations as the steering strength since the learned steering vectors should satisfy the monotonicity described in Assumption 3.4 . Formally, we set $s^{(l)} = \max_{x^{(l)} \in H_{train}^{(l)}} f^{(l)}(\mathbf{x}^{(l)})$.

*Table 1.* The performance of contrastive steering against 12 different fortified LLMs. * means no available direction is found. The best result of each column is in **bold**.

| Method | Llama2-DA | | | Gemma-DA | | | Vicuna-SU | | | Mistral-SU | | | Mistral-RB | | | Llama3-RB | | | Llama3-LAT | | | Llama3-TAR | | | Mistral-CB | | | Llama3-CB | | | R2D2 | | | Llama3-DeRTA | | | Avg. |
|---|---|---|---|---|---|---|---|---|---|---|---|---|---|---|---|---|---|---|---|---|---|---|---|---|---|---|---|---|---|---|---|---|---|---|---|---|---|
| | SRF | HB | SR | SRF | HB | SR | SRF | HB | SR | SRF | HB | SR | SRF | HB | SR | SRF | HB | SR | SRF | HB | SR | SRF | HB | SR | SRF | HB | SR | SRF | HB | SR | SRF | HB | SR | SRF | HB | SR | |
| RepE | 0.01 | 0.00 | 0.00 | 0.01 | 0.00 | 0.01 | 0.01 | 0.00 | 0.02 | 0.02 | 0.01 | 0.03 | 0.04 | 0.01 | 0.04 | 0.04 | 0.01 | 0.07 | 0.00 | 0.00 | 0.00 | 0.00 | 0.00 | 0.00 | 0.02 | 0.02 | 0.00 | 0.01 | 0.02 | 0.02 | 0.04 | 0.04 | 0.00 | 0.01 | 0.04 | 0.00 | 0.02 |
| SCAV | 0.01 | 0.04 | 0.01 | 0.38 | 0.47 | 0.64 | 0.01 | 0.03 | 0.01 | 0.01 | 0.01 | 0.00 | 0.01 | 0.03 | 0.00 | 0.01 | 0.04 | 0.01 | 0.01 | 0.00 | 0.01 | 0.01 | 0.03 | 0.01 | 0.01 | 0.01 | 0.01 | 0.01 | 0.03 | 0.01 | 0.04 | 0.04 | 0.00 | 0.00 | 0.01 | 0.00 | 0.05 |
| RD-A | * | * | * | 0.57 | 0.57 | 0.76 | * | * | * | 0.08 | 0.04 | 0.15 | 0.07 | 0.03 | 0.07 | 0.64 | 0.70 | 0.88 | 0.23 | 0.29 | 0.35 | 0.19 | 0.14 | 0.27 | * | * | * | 0.09 | 0.10 | 0.10 | 0.23 | 0.31 | 0.45 | 0.32 | 0.44 | 0.48 | 0.24 |
| RD-C | * | * | * | 0.37 | 0.50 | 0.55 | * | * | * | 0.04 | 0.03 | 0.08 | 0.23 | 0.25 | 0.34 | 0.26 | 0.41 | 0.44 | 0.09 | 0.13 | 0.14 | 0.15 | 0.16 | 0.27 | * | * | * | 0.28 | 0.38 | 0.35 | 0.18 | 0.25 | 0.39 | 0.07 | 0.13 | 0.15 | 0.18 |
| Angular | 0.01 | 0.05 | 0.00 | 0.23 | 0.18 | 0.26 | 0.01 | 0.00 | 0.01 | 0.04 | 0.02 | 0.07 | 0.07 | 0.01 | 0.12 | 0.06 | 0.03 | 0.08 | 0.10 | 0.12 | 0.14 | 0.25 | 0.22 | 0.40 | 0.02 | 0.02 | 0.01 | 0.01 | 0.01 | 0.02 | 0.25 | 0.35 | 0.50 | 0.25 | 0.37 | 0.46 | 0.13 |
| Ours | **0.57** | **0.86** | **0.85** | **0.69** | **0.73** | **0.88** | **0.60** | **0.75** | **0.88** | **0.46** | **0.57** | **0.77** | **0.58** | **0.63** | **0.85** | **0.71** | **0.86** | **0.98** | **0.71** | **0.82** | **0.98** | **0.32** | **0.24** | **0.50** | **0.72** | **0.81** | **0.95** | **0.70** | **0.83** | **0.91** | **0.31** | **0.41** | **0.64** | **0.61** | **0.78** | **0.91** | **0.70** |

*Table 2.* The performance gain of components proposed for strength tuning.

| Method | Llama2-DA | | | Gemma-DA | | | Vicuna-SU | | | Mistral-SU | | | Mistral-RB | | | Llama3-RB | | | Llama3-LAT | | | Llama3-TAR | | | Mistral-CB | | | Llama3-CB | | | R2D2 | | | Llama3-DeRTA | | | Avg. |
|---|---|---|---|---|---|---|---|---|---|---|---|---|---|---|---|---|---|---|---|---|---|---|---|---|---|---|---|---|---|---|---|---|---|---|---|---|---|
| | SRF | HB | SR | SRF | HB | SR | SRF | HB | SR | SRF | HB | SR | SRF | HB | SR | SRF | HB | SR | SRF | HB | SR | SRF | HB | SR | SRF | HB | SR | SRF | HB | SR | SRF | HB | SR | SRF | HB | SR | |
| SCAV | 0.01 | 0.04 | 0.01 | 0.38 | 0.47 | 0.64 | 0.01 | 0.03 | 0.01 | 0.01 | 0.01 | 0.00 | 0.01 | 0.03 | 0.00 | 0.01 | 0.04 | 0.01 | 0.01 | 0.03 | 0.01 | 0.01 | 0.01 | 0.01 | 0.01 | 0.01 | 0.01 | 0.01 | 0.03 | 0.01 | 0.04 | 0.04 | 0.00 | 0.00 | 0.01 | 0.00 | 0.05 |
| SCAV+DLA+SAT | 0.02 | 0.01 | 0.01 | 0.31 | 0.44 | 0.44 | 0.01 | 0.00 | 0.01 | 0.02 | 0.03 | 0.01 | 0.01 | 0.01 | 0.00 | 0.01 | 0.00 | 0.00 | 0.02 | 0.05 | 0.01 | 0.01 | 0.01 | 0.01 | 0.01 | 0.01 | 0.00 | 0.01 | 0.03 | 0.01 | 0.01 | 0.03 | 0.01 | 0.01 | 0.02 | 0.00 | 0.04 |
| SCAV+AS | 0.04 | 0.37 | 0.04 | 0.49 | 0.49 | 0.74 | 0.01 | 0.50 | 0.74 | 0.01 | 0.01 | 0.00 | 0.01 | 0.03 | 0.01 | 0.01 | 0.02 | 0.01 | 0.01 | 0.03 | 0.01 | 0.01 | 0.00 | 0.01 | 0.01 | 0.00 | 0.00 | 0.01 | 0.01 | 0.00 | 0.01 | 0.01 | 0.00 | 0.06 | 0.21 | 0.17 | 0.14 |
| SCAV+AS+DLA | 0.06 | 0.49 | 0.10 | 0.49 | 0.49 | 0.70 | 0.45 | 0.55 | 0.75 | 0.22 | 0.21 | 0.44 | 0.18 | 0.31 | 0.34 | 0.17 | 0.14 | 0.30 | 0.57 | 0.78 | 0.90 | 0.10 | 0.05 | 0.20 | 0.08 | 0.09 | 0.14 | 0.02 | 0.05 | 0.02 | 0.10 | 0.11 | 0.25 | 0.06 | 0.21 | 0.17 | 0.29 |
| SCAV+AS+DLA+SAT | 0.02 | 0.29 | 0.01 | 0.47 | 0.44 | 0.75 | 0.60 | 0.76 | 0.85 | 0.38 | 0.44 | 0.60 | 0.31 | 0.44 | 0.60 | 0.61 | 0.73 | 0.90 | 0.56 | 0.69 | 0.89 | 0.09 | 0.05 | 0.24 | 0.30 | 0.27 | 0.54 | 0.10 | 0.26 | 0.18 | 0.20 | 0.18 | 0.43 | 0.36 | 0.60 | 0.74 | 0.44 |

## 4.2. Colosseum

**Comparison between Attacks.** As presented in Table 1, our method demonstrates better effectiveness and robustness compared to others. When attacking Gemma-DA, Llama3-RB, and Llama3-CB, which have already shown vulnerability to other contrastive steering, our method can further raise the harmfulness scores on all metrics. In terms of robustness, our method achieves SR scores of at least 0.5 on 12 LLMs, with most reaching at least 0.8, while others exhibit both SR scores larger than 0.4 and close to 0 across different LLMs. Specifically, RepE achieves all near-zero performance. SCAV has near-zero scores on most LLMs, except for Gemma-DA. According to Figure 3, we believe the reason is that Gemma-DA has activation of a larger magnitude than others, which can tolerate the large-scale steering recommended by Xu et al. (2024). RD exhibits the second-best effectiveness on average, yet also fails to find available direction on several LLMs due to its direction filtering.[6]

**Comparison between LLMs.** Among these fortified LLMs, Llama3-TAR (Tamirisa et al., 2025) exhibits the best robustness. We believe this is because TAR conducts adversarial training over fully tampered parameters, while contrastive steering can be seen as tampering only with partial parameters. As a comparison, Llama3-LAT conducts adversarial training over the same activations as contrastive steering, and thus is more vulnerable than Llama3-TAR. The SU-series counters jailbreaking by unlearning harmful knowledge. Yet, without any target response, our method successfully elicits harmful responses, indicating that the SU-series still contains harmful knowledge. RB- and CB-series defend against jailbreaking by manipulating activations, wherein the CB-series was reported that perturbing activations (i.e., applying RepE) can not bypass it (Zou et al., 2024). However, our method, as well as RD, shows that

perturbing activations can jailbreak CB-series. Bypassing DA-series and DeRTA, which both tackle shallow alignment, indicates that the contrastive steering does not depend on shallow alignment (Qi et al., 2025). R2D2 exhibits the second-best robustness, yet, in Section 4.3.3, we will show that such robustness is induced by degraded capability.

## 4.3. Ablation Study

We analyze the indispensability of the proposed four components in Section 4.3.1 and Section 4.3.2. We also demonstrate the correlation between the capability and the harmfulness and how to deal with poor LLMs in Section 4.3.3.

### 4.3.1. STRENGTH TUNING

Let us use the probes trained on initial contrastive activations to investigate how our strength tuning strategy benefits the probe-based steering.

**Adaptive Strength (AS).** The adaptive strength is essential. Without it, even if the searched direction is ideal, we can hardly tackle oversteering unless we conduct a laborious grid search over $L$ continuous parameters. When setting $s^{(l)} = \sigma^{-1}(99.99\%)$ and using the validation accuracy to determine steer or not, as proposed by Xu et al. (2024), we can only get harmfulness scores near zero, as shown in Table 2 and row 2, Table 1. The only exception is Gemma-DA. The reason is that Gemma-DA's activation magnitude is large such that setting $s^{(l)} = \sigma^{-1}(99.99\%)$ will not cause oversteering. After applying AS, as presented in Table 2, we successfully drag Llama2-DA, Vicuna-SU, Llama3-TAR, and Llama3-DeRTA out of the zero zone, while DLA+SAT alone can not.

**Discarding the Last-layer Activation (DLA).** Steering the last-layer activation is equivalent to applying logit bias. While such steering may somehow utilize the shallow alignment (Qi et al., 2025), jailbreaking the LLM by forcing

---

[6]We explain this in Appendix D.

*Table 3.* The performance gain of components proposed for direction searching. * means no available direction is found.

| Method | Llama2-DA | | | Gemma-DA | | | Vicuna-SU | | | Mistral-SU | | | Mistral-RB | | | Llama3-RB | | | Llama3-LAT | | | Llama3-TAR | | | Mistral-CB | | | Llama3-CB | | | R2D2 | | | Llama3-DeRTA | | | Avg. |
|---|---|---|---|---|---|---|---|---|---|---|---|---|---|---|---|---|---|---|---|---|---|---|---|---|---|---|---|---|---|---|---|---|---|---|---|---|---|
| | SRF | HB | SR | SRF | HB | SR | SRF | HB | SR | SRF | HB | SR | SRF | HB | SR | SRF | HB | SR | SRF | HB | SR | SRF | HB | SR | SRF | HB | SR | SRF | HB | SR | SRF | HB | SR | SRF | HB | SR | |
| RepE+NA | 0.01 | 0.00 | 0.00 | 0.01 | 0.00 | 0.00 | 0.01 | 0.00 | 0.02 | 0.01 | 0.00 | 0.05 | 0.05 | 0.00 | 0.07 | 0.03 | 0.03 | 0.09 | 0.00 | 0.00 | 0.00 | 0.08 | 0.10 | 0.14 | 0.01 | 0.01 | 0.02 | 0.00 | 0.00 | 0.00 | 0.04 | 0.06 | 0.03 | 0.00 | 0.01 | 0.00 | 0.03 |
| SCAV+NA | 0.03 | 0.12 | 0.04 | 0.28 | 0.30 | 0.30 | 0.03 | 0.25 | 0.04 | 0.01 | 0.03 | 0.01 | 0.01 | 0.03 | 0.01 | 0.01 | 0.03 | 0.01 | 0.06 | 0.06 | 0.14 | 0.01 | 0.03 | 0.01 | 0.01 | 0.00 | 0.01 | 0.01 | 0.05 | 0.01 | | | | | | | 0.06 |
| RD-A+NA | * | * | * | 0.65 | 0.63 | 0.86 | * | * | * | 0.12 | 0.09 | 0.19 | 0.26 | 0.21 | 0.37 | 0.68 | 0.75 | 0.91 | 0.19 | 0.22 | 0.26 | * | * | * | * | * | * | 0.13 | 0.17 | 0.18 | 0.24 | 0.31 | 0.42 | 0.09 | 0.14 | 0.19 | 0.23 |
| RD-C+NA | * | * | * | 0.58 | 0.69 | 0.76 | * | * | * | 0.04 | 0.03 | 0.07 | 0.28 | 0.29 | 0.41 | 0.25 | 0.38 | 0.40 | 0.12 | 0.13 | 0.18 | * | * | * | * | * | * | 0.11 | 0.22 | 0.17 | 0.29 | 0.38 | 0.49 | 0.05 | 0.10 | 0.12 | 0.18 |
| Angular+NA | 0.14 | 0.14 | 0.22 | 0.32 | 0.31 | 0.43 | 0.01 | 0.00 | 0.01 | 0.14 | 0.10 | 0.27 | 0.15 | 0.08 | 0.25 | 0.03 | 0.03 | 0.05 | 0.19 | 0.23 | 0.30 | 0.27 | 0.22 | 0.42 | 0.03 | 0.03 | 0.03 | 0.02 | 0.02 | 0.02 | 0.22 | 0.27 | 0.44 | 0.38 | 0.52 | 0.59 | 0.19 |
| SCAV+AS+DLA+SAT+NA | 0.16 | 0.41 | 0.24 | 0.55 | 0.56 | 0.83 | 0.54 | 0.69 | 0.82 | 0.23 | 0.28 | 0.40 | 0.33 | 0.31 | 0.64 | 0.57 | 0.69 | 0.86 | 0.56 | 0.68 | 0.89 | 0.12 | 0.05 | 0.22 | 0.28 | 0.26 | 0.55 | 0.46 | 0.50 | 0.79 | 0.16 | 0.23 | 0.34 | 0.33 | 0.47 | 0.63 | 0.46 |
| SCAV+AS+DLA+SAT+AR | 0.57 | 0.86 | 0.85 | 0.69 | 0.73 | 0.88 | 0.60 | 0.75 | 0.88 | 0.46 | 0.57 | 0.77 | 0.58 | 0.63 | 0.85 | 0.71 | 0.86 | 0.98 | 0.71 | 0.82 | 0.98 | 0.32 | 0.24 | 0.50 | 0.72 | 0.81 | 0.95 | 0.70 | 0.83 | 0.91 | 0.31 | 0.41 | 0.64 | 0.61 | 0.78 | 0.91 | 0.70 |

it to start with a harmful tone, such steering has no benefits to coherence. In Table 2, we can find that discarding the last-layer activation further improves the harmfulness scores.

**Steering All Tokens (SAT).** As mentioned in Section 3.3.3, the probe-based steering can be viewed as a form of parameter-efficient adapter and should be applied to all token positions if so. As shown in Table 2, SAT improves performance by 15% on average. Notably, on the CB- and the RB-series, it raises harmfulness scores by a large margin. We believe the reason is that CB and RB manipulate activations of prompt token positions during their training. If only response tokens are steered, the overall generation remains based on manipulated prompt tokens, resulting in unsuccessful steering.

### 4.3.2. MODEL EXTRACTION

So far, we have shown how AS, DLA, and SAT improve and simplify probe-based steering. Yet, with biased directions, the adaptive logit target may correspond to biased behavior. Below, we investigate how our model extraction view benefits direction searching.

**Naive Augmentation (NA).** Adding more contrastive prompts is the simplest approach to sample more annotated activations. We use all 871 harmful prompts and the first 871 benign prompts from Arditi et al. (2024) to extract activations, resulting in 1,742 contrastive activations, which is more than our 20-iteration adaptive retraining has (1,100 at most). In Table 3, we can find that NA only marginally improves the average scores by at most 6%. This result aligns with our insight that the coupled direction induced by contrastive prompts hinders the effectiveness of NA.

**Our Adaptive Retraining (AR).** As shown in Table 3, our AR improves probe-based steering by 26% on average. These results demonstrate AR's competitive effectiveness and data efficiency. We also present the SRF scores on the validation set across iterations in Figure 4. We can observe that the performance of the probe-based steering improves over iterations and fluctuates within a certain range when the AR is converged. Such a tendency is obvious on LLMs other than the SU-series. We hypothesize that, under our default setting, AR may have already converged on the SU-series,

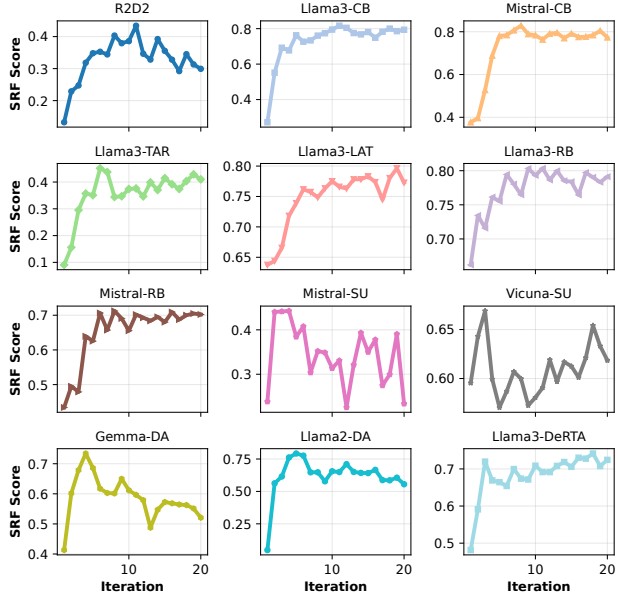

*Figure 4.* The SRF Score of steered LLMs across iterations on the validation set.

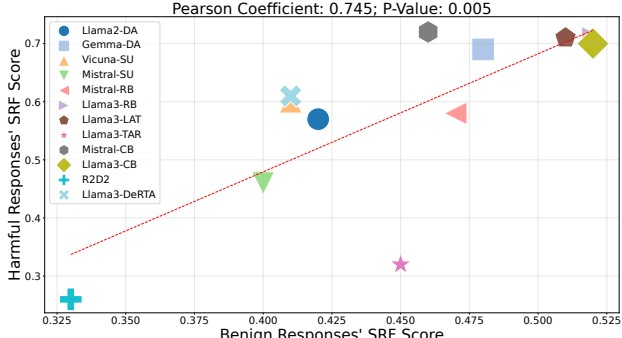

*Figure 5.* Correlation between benign responses' SRF score and harmful responses' SRF score across different LLMs.

leading to such fluctuation.

### 4.3.3. CAPABILITY AND SAFETY

In Table 1, R2D2 exhibits the second-best harmlessness. Yet, we find that such harmlessness is induced by degraded overall helpfulness. Using SRF to judge helpfulness, we evaluate the response quality of all 12 LLMs. In Figure 5, we can find a clear positive correlation between the help-

Table 4. R2D2's harmfulness scores.

| Method | R2D2 | | |
| --- | --- | --- | --- |
| | SRF | HB | SR |
| Ours | 0.31 | 0.41 | 0.64 |
| Ours+Filter | 0.38 | 0.51 | 0.65 |
| Ours+Response | 0.53 | 0.67 | 0.69 |
| Ours+Filter+Response | 0.50 | 0.64 | 0.58 |
| Ours+Filter+Response+(0.05, 0.8) | 0.74 | 0.87 | 0.81 |

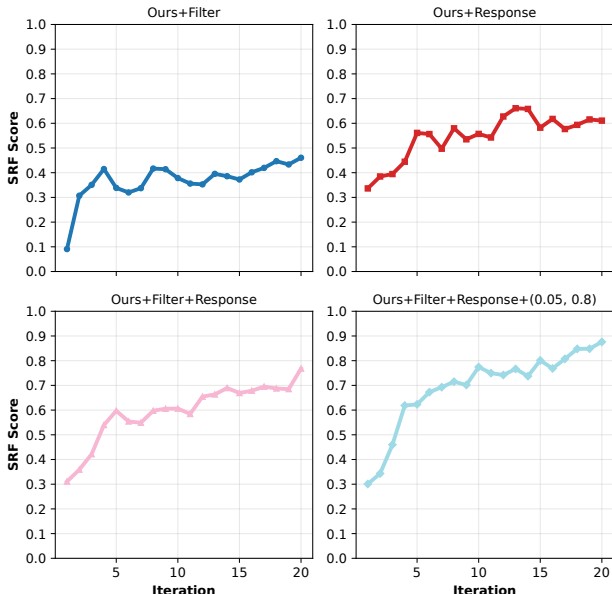

Figure 6. The SRF Score of steered R2D2 across iterations on the validation set.

fulness of harmful responses and benign responses. Notably, R2D2 has the worst helpfulness (We manually check R2D2's responses. Even if the prompt is benign, it still tends to respond with "I am not able to do..."). That is to say, if the LLM can not properly follow benign prompts, it can thus hardly follow harmful ones, even if it is willing to follow.

While in practice, one can ignore poor LLMs and choose to jailbreak advanced LLMs for high-quality responses, it remains interesting to consider how to deal with LLMs that consistently exhibit poor attitudes. Let us take R2D2 as an example. First, observing the clear positive correlation between the helpfulness of harmful responses and benign responses, we propose to filter out benign prompts that R2D2 can not (or refuses to) follow, i.e., those with an SRF score lower than 0.5. Second, assuming response tokens contain more information than the single token preceding the first response token, we collect and average activations from response tokens for direction searching. Third, we

lift the high threshold from 0.6 to 0.8, which will exclude samples that SRF is less certain about.

In Table 4, we can find that both using activations from response tokens and filtering contrastive prompts can improve our method. In Figure 6, we can find that including response tokens unleashes the upper limit of our method's performance on R2D2 and that lifting the high threshold and filtering enables our methods to ascend more quickly and stably.

It is intuitive that filtering contrastive prompts can promote contrastive steering, since contrastive prompts must be contrastive. As for using activations from response tokens, we believe activations from response tokens contain useful information(e.g., linguistic style, reasoning style, etc.), benefiting LLMs that can hardly follow benign instructions (a basic capability for aligned LLMs). We manually check the responses of R2D2 with filtered data and response activations. We find that the response style shifts from "I do not have the capability" to "Step 1. ... Step 2. ...", a style preferred by most jailbreaking (Gong et al., 2025).

## 5. Conclusion

By utilizing the statistics of contrastive activations and adaptive retraining, we successfully improved the robustness and effectiveness of probe-based steering. We demonstrated that our method can reveal the vulnerability of LLMs that prior steering did not reveal and can improve the harmfulness of LLMs that have already shown vulnerability to prior steering. We hope that our method can benefit the robustness evaluation (red-teaming), which is the foundation of developing truly reliable defenses.

## Acknowledgements

The work is supported by National Natural Science Foundation of China (12326618) and the Project of Guangdong Provincial Key Laboratory of Information Security Technology (Grant No. 2023B1212060026).

## Impact Statement

While the proposed algorithm can be directly applied to jailbreak open-source LLMs, we believe it also enhances the effectiveness and robustness of red-teaming, an essential step toward building reliable defenses. Moreover, since the algorithm is evaluated in terms of eliciting a faithful persona, we believe that, assuming such persona control is not a special case, it could also be extended to control other LLM personas for beneficial applications.

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

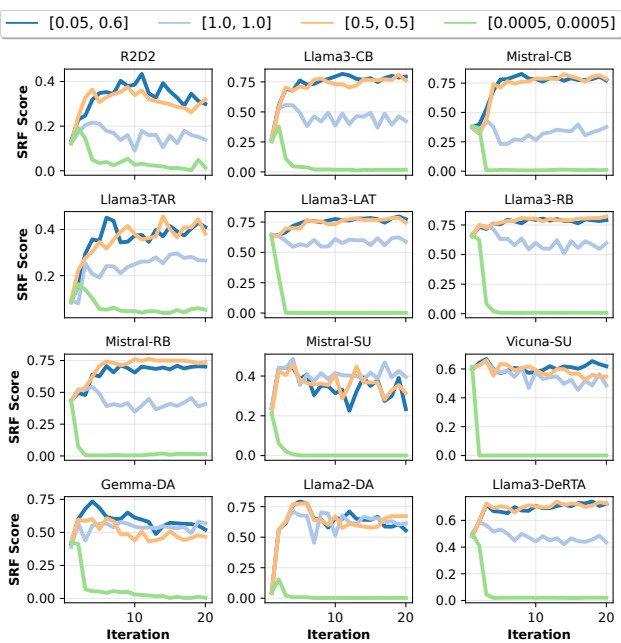

*Figure 7.* The SRF Score of the validation set across iterations under different SRF thresholds.

## A. Annotator Threshold Tuning

Currently, the most exhausting part of our method is the annotator. The annotator serves as a proxy for ideal probes but is inevitably noisy in practice. For the SRF we use, a faithless but detailed disclaimer may be scored 0.4. In contrast, the truncated beginning of a faithful but lengthy response may be scored only 0.2. This forces us to set a low threshold and a high threshold that are far apart, which helps avoid the ambiguous score interval.

We note that setting a uni-threshold, which ignores the ambiguous score interval, may achieve acceptable performance due to the data distribution. As shown in Figure 7, setting $(0.5, 0.5)$ achieves comparable performance to our default bi-threshold, $(0.05, 0.6)$. Yet, in Figure 8, we can find that most violin plots are shaped like long-necked funnels or hourglasses. Comparing Figure 9 and Figure 7, we can find that setting $(1.0, 1.0)$, which classifies all samples as faithless, approximates $(0.05, 0.6)$ only on Mistral-SU because most of Mistral-SU's responses are indeed faithless. Due to these polarized distributions, even if the SRF is set as a uni-threshold classifier, its classification results approximate those of a bi-threshold SRF. Yet, when the distribution of SRF Scores does not favor the uni-threshold SRF, as shown in Figure 10 and Figure 7, Algorithm 1 collapses.

While bi-threshold can improve SRF's accuracy and thus Algorithm 1's robustness, it may discard a considerable number of generated responses, leading to waste and inefficiency. Consequently, our universal setting, 0.05 and 0.6, is

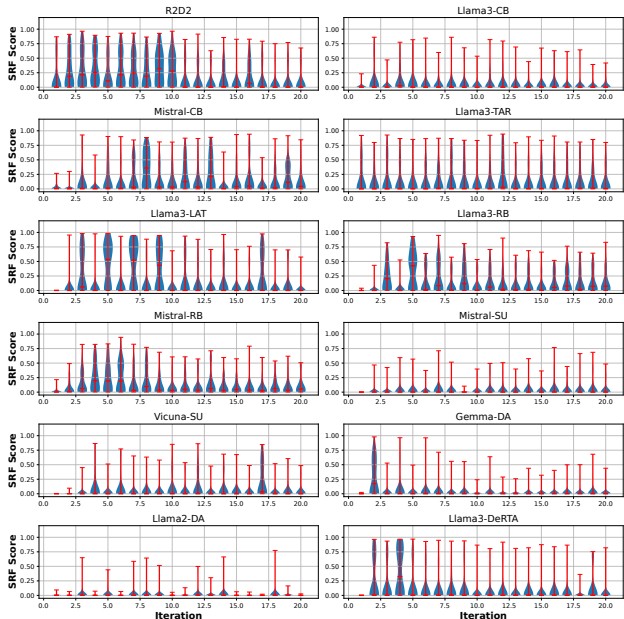

*Figure 8.* The violin plot of sampled responses' SRF Score across iterations. The SRF's threshold is 0.5.

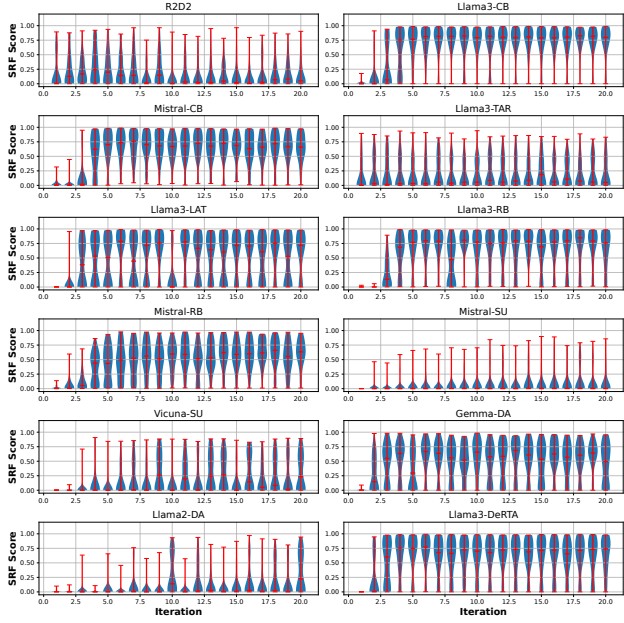

*Figure 9.* The violin plot of sampled responses' SRF Score across iterations. The SRF's threshold is 1.0.

the result of a trade-off between robustness and efficiency. Note that our default settings, 0.05 and 0.6, may not be the optimal settings for all LLMs. As shown in Table 5, while the average harmfulness score is stable, the best SRF thresholds for each LLM vary. Notably, on Gemma-DA, the range of harmfulness scores across different thresholds can reach up to 20%. We hypothesize the reason is that different LLMs have distinct linguistic styles, and because

*Table 5.* The performance of our method with different SRF thresholds. The best result for each LLM is in **bold**.

| Method | Llama2-DA SRF | HB | SR | Gemma-DA SRF | HB | SR | Vicuna-SU SRF | HB | SR | Mistral-SU SRF | HB | SR | Mistral-RB SRF | HB | SR | Llama3-RB SRF | HB | SR | Llama3-LAT SRF | HB | SR | Llama3-TAR SRF | HB | SR | Mistral-CB SRF | HB | SR | Llama3-CB SRF | HB | SR | R2D2 SRF | HB | SR | Llama3-DeRTA SRF | HB | SR | Avg. |
|---|---|---|---|---|---|---|---|---|---|---|---|---|---|---|---|---|---|---|---|---|---|---|---|---|---|---|---|---|---|---|---|---|---|---|---|---|---|
| Ours (0.05, 0.8) | 0.54 | 0.87 | 0.78 | **0.74** | **0.77** | **0.95** | **0.62** | **0.75** | **0.89** | 0.45 | 0.50 | 0.76 | 0.60 | 0.62 | 0.89 | 0.72 | 0.83 | 0.97 | **0.73** | **0.85** | **0.98** | 0.33 | 0.25 | 0.49 | 0.70 | 0.74 | 0.96 | 0.68 | 0.82 | 0.91 | 0.31 | 0.41 | 0.61 | 0.58 | 0.74 | 0.94 | 0.70 |
| Ours (0.05, 0.4) | 0.58 | 0.87 | 0.82 | 0.59 | 0.57 | 0.77 | 0.61 | 0.71 | 0.88 | **0.49** | **0.58** | **0.80** | **0.67** | **0.76** | **0.91** | 0.72 | 0.82 | 0.97 | 0.70 | 0.79 | 0.97 | **0.38** | **0.29** | **0.58** | 0.68 | 0.80 | 0.96 | 0.67 | 0.78 | 0.90 | **0.31** | **0.43** | **0.68** | **0.64** | **0.83** | **0.95** | 0.71 |
| Ours (0.05, 0.6) | 0.57 | 0.86 | 0.85 | 0.69 | 0.73 | 0.88 | 0.60 | 0.75 | 0.88 | 0.46 | 0.57 | 0.77 | 0.58 | 0.63 | 0.85 | **0.71** | **0.86** | **0.98** | 0.71 | 0.82 | 0.98 | 0.32 | 0.24 | 0.50 | **0.72** | **0.81** | **0.95** | 0.70 | 0.83 | 0.91 | 0.31 | 0.41 | 0.64 | 0.61 | 0.78 | 0.91 | 0.70 |
| Ours (0.1, 0.6) | **0.65** | **0.89** | **0.90** | 0.60 | 0.56 | 0.75 | 0.61 | 0.74 | 0.89 | 0.46 | 0.54 | 0.80 | 0.64 | 0.74 | 0.89 | 0.72 | 0.84 | 0.97 | 0.71 | 0.82 | 0.97 | 0.35 | 0.23 | 0.54 | 0.67 | 0.74 | 0.93 | **0.69** | **0.87** | **0.95** | 0.30 | 0.38 | 0.63 | 0.62 | 0.81 | 0.95 | 0.70 |
| Ours (0.01, 0.6) | 0.66 | 0.85 | 0.86 | 0.67 | 0.63 | 0.85 | 0.61 | 0.73 | 0.91 | 0.44 | 0.52 | 0.75 | 0.61 | 0.75 | 0.91 | 0.67 | 0.80 | 0.96 | 0.71 | 0.83 | 0.97 | 0.33 | 0.25 | 0.48 | 0.69 | 0.72 | 0.94 | 0.71 | 0.81 | 0.94 | 0.28 | 0.34 | 0.65 | 0.64 | 0.81 | 0.94 | 0.70 |

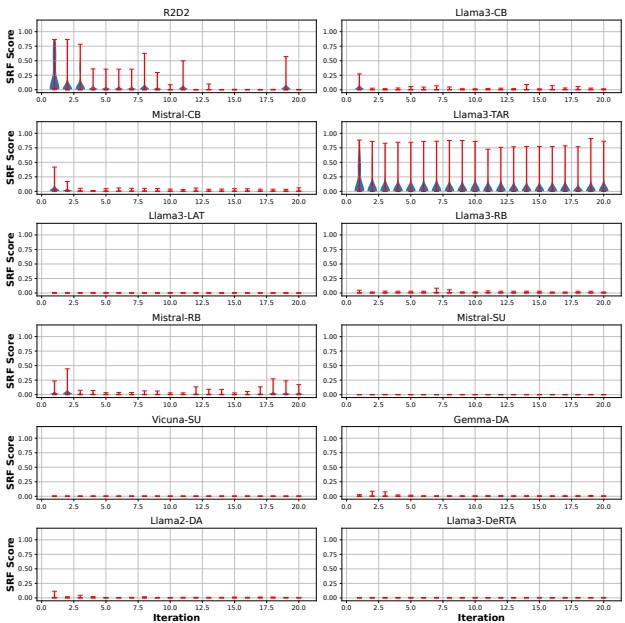

*Figure 10.* The violin plot of sampled responses' SRF Score across iterations. The SRF's threshold is 0.0005.

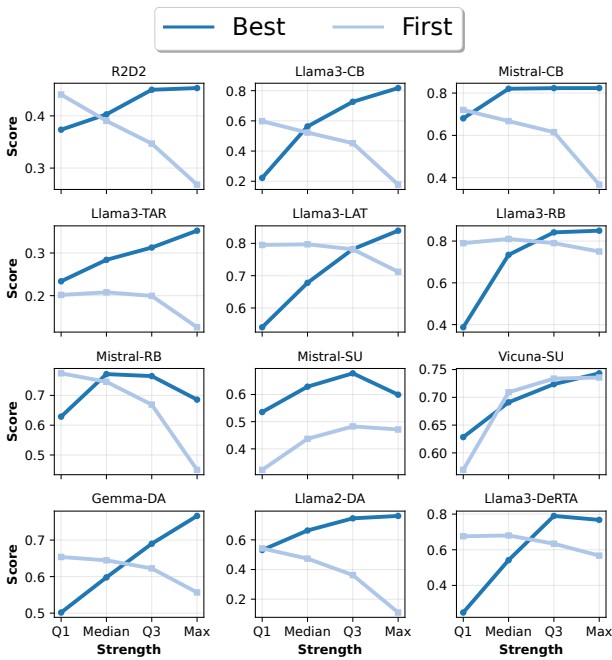

*Figure 11.* The harmfulness score of LLMs with different steering strengths.

SRF may prefer a particular style, which makes the optimal SRF threshold for each LLM vary.

Using closed-source LLMs as a rubric annotator might reduce the need for exhaustive threshold tuning, but closed-source LLMs often exhibit randomness (He & Lab, 2025), making them even harder to control and making SRF with thresholds the best annotator we know of in terms of efficiency, accuracy, and controllability. Currently, we do not bother and do not recommend conducting a grid search on these thresholds because tuning thresholds will not improve the underlying annotator but will instead reduce misclassified samples at the expense of efficiency. Instead, it is more worthwhile to build a stronger new annotator.

Note that the exhaustive threshold tuning is not an inherent limitation of our method. We believe that powerful annotators in the future (or expensive human annotators at present) will trivially free our method from threshold tuning.

## B. Monotonicity

Recall Assumption 3.4 that a learned steering vector sequence $V$ should satisfy that if $m_\theta(\Delta\theta)$ is $(V, H, S)$-Similar, then $\mathbb{I}_\beta(m_\theta(\Delta\theta))$ is a monotonically increasing function of $S$ on the interval $[\underline{S}, \bar{S}]$. Satisfying such monotonicity brings two benefits for implementation. First, we can set the steering strength to the maximum known logit (i.e., $s^{(l)} = \max_{x^{(l)} \in H_{train}^{(l)}} f^{(l)}(\mathbf{x}^{(l)})$) to gain the best steering performance, which eliminates the need for strength tuning. Second, the steering vector can be applied to monitor the concept within the LLM (Chen et al., 2025; Arditi et al., 2024).

Throughout our paper, unless otherwise specified, we set the steering strength to the maximum known logit during the inference because we require our method to be tuning-free. Such simplicity can be achieved because Algorithm 1 improves the learned steering vector's monotonicity. We calculate the logits of all known faithful activations and derive the corresponding lower quartile (Q1), median, upper quartile (Q3), and maximum as steering strengths. The harmfulness score is calculated by averaging the scores of SRF, HB, and SR. As shown in Figure 11, when the probe is trained on vanilla contrastive prompts' activations (labeled "First"), 10 out of 12 steered LLMs achieved the highest

*Table 6.* The performance of contrastive steering against 7 different general-purpose LLMs. * means no available direction is found. The best result of each column is in **bold**.

| Method | Qwen3-4B-Instruct-2507 | | | Qwen2.5-14B-Instruct | | | Llama-3-8B-Instruct | | | Mistral-7B-Instruct-v0.2 | | | Llama-2-7b-chat | | | Vicuna-7b-v1.5 | | | Gemma-2-9b-it | | | Avg. |
| --- | --- | --- | --- | --- | --- | --- | --- | --- | --- | --- | --- | --- | --- | --- | --- | --- | --- | --- | --- | --- | --- | --- |
| | SRF | HB | SR | SRF | HB | SR | SRF | HB | SR | SRF | HB | SR | SRF | HB | SR | SRF | HB | SR | SRF | HB | SR | |
| RepE | 0.02 | 0.00 | 0.00 | 0.08 | 0.03 | 0.07 | 0.01 | 0.05 | 0.00 | 0.02 | 0.02 | 0.03 | 0.03 | 0.02 | 0.02 | 0.11 | 0.15 | 0.21 | 0.01 | 0.00 | 0.00 | 0.04 |
| SCAV | 0.56 | **0.81** | 0.84 | **0.70** | **0.88** | **0.96** | 0.01 | 0.03 | 0.01 | 0.01 | 0.02 | 0.00 | 0.01 | 0.00 | 0.01 | 0.01 | 0.00 | 0.00 | 0.30 | 0.41 | 0.57 | 0.29 |
| RD-A | * | * | * | * | * | * | * | * | * | 0.69 | 0.79 | 0.92 | * | * | * | * | * | * | 0.66 | 0.71 | 0.85 | 0.22 |
| RD-C | * | * | * | * | * | * | * | * | * | 0.54 | 0.77 | 0.87 | * | * | * | * | * | * | **0.71** | **0.86** | **0.94** | 0.22 |
| Angular | 0.55 | 0.63 | 0.80 | 0.63 | 0.67 | 0.91 | 0.67 | 0.79 | 0.93 | 0.68 | 0.81 | 0.93 | 0.72 | 0.85 | 0.91 | 0.58 | 0.75 | 0.88 | 0.02 | 0.22 | 0.01 | 0.66 |
| Ours | **0.71** | 0.80 | **0.96** | 0.61 | 0.69 | 0.91 | **0.72** | **0.84** | **0.98** | **0.77** | **0.95** | **0.97** | **0.73** | **0.89** | **0.94** | **0.59** | **0.81** | **0.90** | 0.65 | 0.64 | 0.83 | **0.80** |

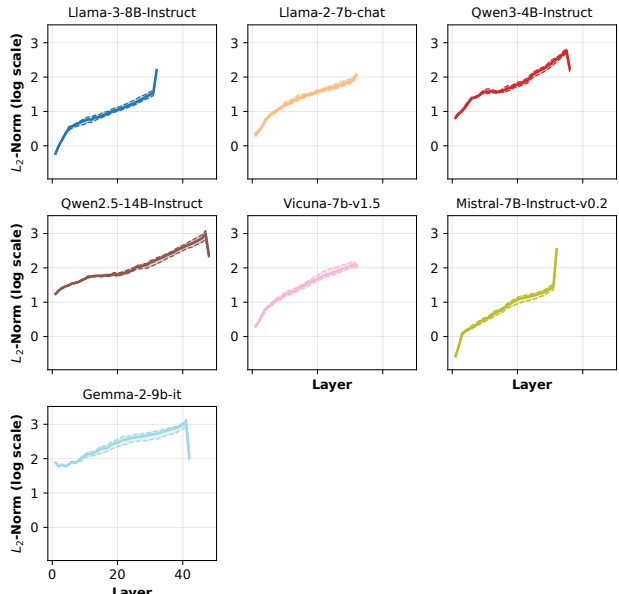

*Figure 12.* $L_2$-norm of activation across layers. The line represents the mean norm, and the shaded area indicates the max and min values of the norm for 100 different activation.

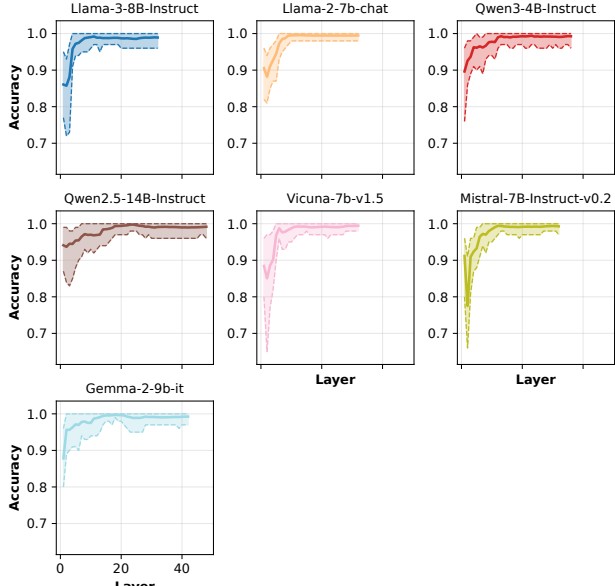

*Figure 13.* Linear probes' accuracy on validation activations across layers. The line represents the mean accuracy, and the shaded area indicates the max and min values of accuracy over 50 random samplings.

harmfulness score when the steering strength is Q1. As a comparison, after executing Algorithm 1 (labeled "Best"), most LLMs reached the highest harmfulness score at the Max and Q3 positions. These results indicate that our probes assign high scores to faithful activations and, thus, better satisfy the monotonicity required by Assumption 3.4.

## C. Results on General-purpose LLMs

As today's investors are more focused on the core capabilities of LLMs (currently demonstrated in areas such as math and coding), mainstream open-source LLMs generally lack specialized defenses against jailbreaks. In Table 6, we can find that, while our method still achieves the highest average harmfulness score, showing robustness, others also have higher harmfulness scores than those of fortified LLMs. In Figure 12, we can find that the Qwen-series's and the Gemma-2-9b-it's activations tend to have large magnitude such that SCAV's default logit target will not induce oversteering, which explains why SCAV performs well on the

Qwen-series and Gemma-2-9b-it.

We believe an attack paper should reveal unrevealed vulnerabilities and thus leave the some-other-trivial-advancement (SOTA) on known vulnerabilities in the appendix.

## D. Why Does RD Fail?

One notable phenomenon throughout our experiment is that RD (Arditi et al., 2024) sometimes fails to find available directions for steering. The problem lies in RD's filtering process.

Given a $L$-layer LLM, whose chat template has $I$ tokens after the instruction, RD collects $I \times L$ candidate directions. During the filtering process, if a direction $V$ fails to induce refusal or significantly alters the LLM's behavior on benign prompts, $V$ will be filtered out. Obviously, once all candidates are discarded, RD fails. In the original paper (Arditi et al., 2024), Arditi et al. (2024) reported that RD can run on

*Table 7.* The performance of our method with different steering strengths and token positions for direction searching. The best result for each LLM is in **bold**.

| Method | Llama2-DA | | | Gemma-DA | | | Vicuna-SU | | | Mistral-SU | | | Mistral-RB | | | Llama3-RB | | | Llama3-LAT | | | Llama3-TAR | | | Mistral-CB | | | Llama3-CB | | | R2D2 | | | Llama3-DeRTA | | | Avg. |
|---|---|---|---|---|---|---|---|---|---|---|---|---|---|---|---|---|---|---|---|---|---|---|---|---|---|---|---|---|---|---|---|---|---|---|---|---|---|
| | SRF | HB | SR | SRF | HB | SR | SRF | HB | SR | SRF | HB | SR | SRF | HB | SR | SRF | HB | SR | SRF | HB | SR | SRF | HB | SR | SRF | HB | SR | SRF | HB | SR | SRF | HB | SR | SRF | HB | SR | |
| Default | **0.57** | **0.86** | **0.85** | 0.69 | 0.73 | 0.88 | **0.60** | **0.75** | **0.88** | 0.46 | 0.57 | 0.77 | 0.58 | 0.63 | 0.85 | 0.71 | 0.86 | 0.98 | 0.71 | 0.82 | 0.98 | 0.32 | 0.24 | 0.50 | 0.72 | 0.81 | 0.95 | 0.70 | 0.83 | 0.91 | 0.31 | 0.41 | 0.64 | 0.61 | 0.78 | 0.91 | 0.70 |
| Default+Med | 0.49 | 0.72 | 0.75 | 0.70 | 0.77 | 0.94 | 0.61 | 0.74 | 0.85 | 0.52 | 0.63 | 0.83 | **0.63** | **0.75** | **0.92** | 0.61 | 0.74 | 0.92 | 0.74 | 0.89 | 0.97 | 0.39 | 0.29 | 0.56 | **0.77** | **0.88** | **0.89** | **0.72** | **0.89** | **0.95** | 0.38 | 0.46 | 0.71 | **0.68** | **0.83** | **0.94** | 0.72 |
| Default+Response | 0.48 | 0.61 | 0.76 | 0.65 | 0.77 | 0.82 | 0.56 | 0.57 | 0.76 | 0.55 | 0.43 | 0.73 | 0.03 | 0.00 | 0.00 | **0.75** | **0.84** | **0.97** | 0.81 | 0.96 | 0.99 | 0.40 | 0.32 | 0.56 | 0.02 | 0.00 | 0.00 | 0.45 | 0.48 | 0.60 | **0.53** | **0.67** | **0.69** | 0.58 | 0.74 | 0.81 | 0.55 |
| Default+Med+Response | 0.40 | 0.50 | 0.61 | **0.74** | **0.88** | **0.96** | 0.58 | 0.58 | 0.81 | **0.66** | **0.64** | **0.78** | 0.03 | 0.00 | 0.01 | 0.33 | 0.36 | 0.51 | 0.81 | 0.92 | 0.97 | **0.52** | **0.51** | **0.65** | 0.02 | 0.00 | 0.00 | 0.66 | 0.76 | 0.88 | 0.46 | 0.60 | 0.75 | 0.48 | 0.55 | 0.66 | 0.54 |

Llama3-8B and Llama2-7B. The only difference in setting between ours and Arditi et al. (2024)'s is that we use the first 100 pairs of contrastive prompts while Arditi et al. (2024) uses 160 pairs. For RD, we set the ratio of the training set to the validation set as 8:2, consistent with Arditi et al. (2024) (we follow the original ratio of the training set to the validation for other methods as well). Such failure may indicate that RD's filtering is not robust and the need for relaxed filtering criteria. We argue that the 100 vs. 160 is not the fundamental reason for RD's failure. In Table 3, we observe that when scaling the contrastive prompts to 871 pairs, RD fails on Llama3-TAR. Yet, when the dataset contains only 100 pairs of contrastive prompts, our default setting, RD, successfully runs on Llama3-TAR.

We find that different steering papers use their own contrastive prompts. To ensure accurate reproduction, it is necessary to use the original paper's dataset. However, we believe that, at a minimum, using the same contrastive prompts for all methods within the same paper is a fundamental control variable. Meanwhile, provided the data meets the basic requirements for contrastive guidance (e.g., harmful prompts induce faithlessness, and harmless prompts induce faithfulness), the contrastive steering's performance should not alter significantly due to variations in the data, which is a fundamental requirement for robust contrastive steering. Therefore, in this paper, we use the same contrastive prompts (though the ratios of training and validation sets may differ) to reproduce all methods.

## E. Why Don't We Sample Benign Prompts' Activations?

In Algorithm 1, one can find that we only sample activations of harmful prompts and may wonder why we do not sample benign prompts' activations. The reasons are as follows.

First, sample benign prompts' activations will make annotation, which has already been the most exhaustive part of our method, even more complex. Take the SRF we use as an example. In Figure 5, we can find that jalibroken responses tend to have higher scores than benign responses. To tackle such bias, we have no choice but to set an extra two thresholds for SRF, which further complicates our method in practice. Even if the annotator can be threshold-free, we still need to carefully address the differences between benign and harmful prompts during the design of such a threshold-free annotator.

Second, the space of faithlessness is larger than the space of faithfulness. Steering from faithlessness toward faithfulness is challenging since one prompt typically corresponds to a limited set of faithful responses. Yet, given an arbitrary prompt, we can easily induce faithless responses, not belonging to the small faithful set, by steering it along a random direction (This is exactly what CB (Zou et al., 2024), a defense, does to harmful activations, which induce incoherent responses). Thus, when we successfully steer faithlessness toward faithfulness, we know that the direction and the strength of steering are both right, and thus the steered sample is meaningful. Yet, when we "successfully" steer faithfulness toward faithlessness, since any random direction can do so, we can hardly say that the steered sample really means something. As for unsuccessfully steered samples, they are all meaningful since they indicate that either the direction or the strength is wrong.

Of course, facing such a generalized definition of "faithlessness", we can redefine "faithlessness": a response is called "faithless" if the response exhibits a similar persona to responses of the LLM facing forbidden prompts. Normally, such a persona is refusal, and we can construct another annotator that detects such "faithlessness" to determine whether we successfully steer benign prompts. However, different LLMs may exhibit distinct personalities when encountering forbidden prompts. For example, CB (Zou et al., 2024) responds incoherently when facing forbidden prompts. Thus, we even have to design different annotators for different LLMs if we insist on steering benign prompts, which will make our method complex and non-general.

In summary, not sampling benign prompts' activations is a compromise made due to the limited capacity of annotators and the asymmetry between faithfulness and faithlessness.

## F. Making Progress under Our Framework.

We are dedicated to designing Algorithm 1 as simpe as possible, enabling readers to focus more on the design rationale rather than tricks. However, this does not mean our algorithm lacks room for improvement. For example, in Section 4.3.3, we demonstrate that, when attacking R2D2, our method can be further improved by including activations of response tokens and applying a stricter annotator.

*Table 8.* The performance of our method against reasoning models and multimodal defenses. * means the model does not support image inputs.

| Method | Qwen3-4B-Thinking | | | Llava-CB | | | GLM-4.6V-Flash | | |
|---|---|---|---|---|---|---|---|---|---|
| | SRF | HB | SR | SRF | HB | SR | SRF | HB | SR |
| No Attack | 0.19 | 0.03 | 0.04 | 0.01 | 0.01 | 0.00 | 0.26 | 0.28 | 0.29 |
| Ours | 0.67 | 0.94 | 0.97 | 0.66 | 0.88 | 0.96 | 0.71 | 0.85 | 0.97 |
| PGD+Ours | * | * | * | 0.66 | 0.81 | 0.92 | 0.80 | 0.97 | 0.98 |

Another component that can be improved is the steering strength $S$ in Algorithm 1. We set $s^{(l)}$ to 0 by default because this setting is widely adopted in the adaptive retraining or active learning literature, known as uncertainty sampling. Yet, we note that both model extraction and active learning are fields that have developed over a long period of time. Beyond uncertainty sampling, these areas have accumulated a wide array of different sampling strategies, most of which claim to be superior to the basic uncertainty sampling (otherwise they can hardly be published).

We made a preliminary attempt. We changed the sampling intensity from 0 to the median of faithful activations' logits. Such a change shifts the sampling strategy from uncertainty sampling to certainty sampling, where the sampled activations are likely to correspond to a faithful persona from the perspective of current probes. In Table 7, we observe that certainty sampling improves the average harmfulness score by 2% and notably enhances performance on Llama3-TAR when we also utilize activations from response tokens. These preliminary results suggest that replacing our default uncertainty sampling with alternative sampling methods is likely to yield better results.

To conclude, we believe that making progress under our framework lies in enhancing the accuracy of the annotator, identifying token positions highly correlated with behavior, and designing a more effective sampling strategy.

## G. Application to Reasoning Models

All defenses that we evaluated in the main paper are developed based on instructed models. Recently, the training scheme of LLMs have shifted from instruction to reasoning. Numerous studies have claimed that reasoning can improve LLMs' adversarial robustness because of its inference-time scaling (Zaremba et al., 2025). We employ our method to test the reasoning model's adversarial robustness against steering. We discard CoT and only keep the answers for calculating harmfulness scores because the CoT is usually found to be helpful for malicious use even if the model is aware of the malicious intent (Wu et al., 2025). Keeping CoT may lead to overestimated harmfulness.

In Table 8, we can find that both Qwen3-4B-Thinking and GLM-4.6V-Flash exhibit extremely high harmfulness scores, demonstrating even worse robustness than the 12 defenses we evaluated. We believe the reasons are as follows. First, steering is essentially an adapter of the victim model. If the so-called safety scales with the CoT, then the effect of steering will also be scaled with CoT. Second, the reasoning model's responses are usually more coherent and helpful than the instructed model's. Since, SRF, HB, and SR all favor harmful and helpful responses, it is trivial that these reasoning models exhibit higher harmfulness scores. These results also, again, support our claim in Section 4.3.3: With robustness unchanged, the model's harmfulness is positively correlated with its usefulness.

## H. Boosting with Prompt-level Jailbreaking

Steering can not be applied if the victim model accepts inputs only. Yet, more than direct control, steering can also be utilized to build up loss functions for optimizing adversarial examples (AEs) (Xu et al., 2024; Huang et al., 2025) that can also inducing jailbreaking. We consider Llava-CB (Zou et al., 2024) and GLM-4.6V-Flash (Hong et al., 2025), an instructed multimodal LLM and a reasoning multimodal LLM, respectively. We consider multimodal LLMs only because visual adversarial examples can be optimized with the powerful gradient descent (e.g., PGD (Madry et al., 2018)) while text space optimization is still too weak to tell whether the loss function truly works.

We first steer the underlying text model with our method and collect activations of the 50 harmful prompts that we used for adaptive retraining. Then, we optimize the image adversarial examples to align the activations of the AE-inputted MLLM and the steered MLLM. The range of AE is limited to $[0, 255]$. Such a procedure can be seen as using visual-prompt-tuning to distill the steered MLLM. After optimizing the AE, we test it with 200 held-out harmful prompts from HarmBench and StrongReject.

In the last row of Table 8, we can find that the performance of AEs approximates or even surpasses the steering. Llava-CB, which is claimed to be robust against AEs that maximizing the probability of compromised prefixes (e.g., "Sure, here is...") (Zou et al., 2024), is bypassed. This result shows

*Table 9.* The performance of contrastive steering against 3 AdaSteer LLMs. **\*** means no available direction is found. The best result of each column is in **bold**.

| Method | Llama-3.1-8B-AdaSteer | | | Gemma-2-9b-AdaSteer | | | Qwen2.5-7B-AdaSteer | | |
| --- | --- | --- | --- | --- | --- | --- | --- | --- | --- |
| | SRF | HB | SR | SRF | HB | SR | SRF | HB | SR |
| RepE | 0.11 | 0.21 | 0.21 | 0.01 | 0.00 | 0.01 | 0.04 | 0.05 | 0.06 |
| SCAV | 0.01 | 0.00 | 0.00 | 0.30 | 0.47 | 0.53 | 0.23 | 0.51 | 0.37 |
| RD-A | * | * | * | 0.33 | 0.35 | 0.42 | * | * | * |
| RD-C | * | * | * | 0.18 | 0.26 | 0.29 | * | * | * |
| Angular | 0.62 | 0.73 | 0.82 | 0.02 | 0.01 | 0.00 | **0.65** | **0.75** | **0.84** |
| Ours | **0.65** | **0.79** | **0.88** | **0.50** | **0.51** | **0.73** | 0.62 | 0.73 | 0.82 |

that maximizing the probability of compromised prefixes is not strong and adaptive enough for evaluating the adversarial robustness for jailbreaking, which may lead to a false sense of security (Athalye et al., 2018; Nasr et al., 2025). As for GLM-4.6V-Flash, inference-time scaling does not help even if no steering scales with the CoT. The AE alone is enough for breaking the so-called robustness boosting by CoT.

Given the results above, we believe steering is meaningful even if the direct steering is not available because the steering can be a bridge (or proxy) to the security of LLMs. With such a bridge and the increasingly incorporated continuous multimodal inputs[7], we can, to some extent, shift the adversarial evaluation of M/LLM's security from the heuristic and manual track back to the automatic track, which is the basis of reliable and scalable adversarial evaluation.

## I. AdaSteer

During the review, reviewer YZiS urged us to include AdaSteer[8], a defense based on steering. We broke it. We also claimed that, when facing steering attacks, the steering-based defense can hardly be stronger than other fine-tuning based defenses because the steering is essentially an adapter. Results are in Table 9. We can find that, while our method achieves non-trivial harmfulness scores, some of the other steering attacks also do well.

## J. Limitations and Future Work

**Efficiency in practice.** The main limitation of our method is its time cost, which is mainly caused by generating responses for annotation. During the direction searching, we set the maximum response length to 256 such that SRF can judge responses accurately enough while the time induced

*Table 10.* Settings of baselines.

| Baselines | Train:Val | Strength |
| --- | --- | --- |
| RepE | 1:0 | 1.00 |
| SCAV | 1:9 | 1e-4 |
| RD-A | 4:1 | - |
| RD-C | 4:1 | 1.0 |
| Angular | 1:0 | 180° |

by generation is acceptable. We believe such a limitation can be trivially mitigated by a more advanced GPU, a more efficient inference framework, and more powerful annotators that can judge activations with shorter truncated responses.

**Application to Controlling Other Personas or Behaviors.** This paper focuses on jailbreaking, wherein the faithfulness is the persona we are concerned with. If such a persona is not special, we believe our method can be utilized to control other personas or behaviors. Yet, as we identified, LCA determines the theoretical feasibility, and, to date, the annotator's reliability determines the practical feasibility.

**Further Improvement.** Rather than a specific algorithm, what we propose in this paper is a model-extraction-inspired steering framework. The three main components of this framework are the annotator, the sampler, and the activation extractor. In Appendix A and Table 4, we show that SRF's thresholds can notably influence our method on certain LLMs. In Appendix F, we demonstrate that using a different sample strategy and activations from different token positions can also improve our method on certain LLMs. These results indicate that the main efforts to improve the performance of our method should focus on: employing better annotators, designing adaptive samplers, and developing adaptive activation extractors.

---

[7]Defenses may hide behind the barrier relying on the hard-to-optimize text space. Yet, such barrier may be broken by future strong text-space discrete optimization.

[8]https://github.com/MuyuenLP/AdaSteer/tree/master

