# OpenReview forum: "Adaptive Probe-based Steering for Robust LLM Jailbreaking"
_ICML.cc/2026/Conference — ICML 2026 regular_

### Official Review · Reviewer_6erV · 2026-03-05

**Soundness:** 3
**Presentation:** 2
**Significance:** 3
**Originality:** 3
**Overall Recommendation:** 3
**Confidence:** 2

**Summary:**

This paper introduces an Adaptive Probe-based Steering framework designed to enhance the robustness of jailbreak attacks on Large Language Models (LLMs). The authors leverage model extraction principles to learn steering vectors and propose an automated strength-tuning mechanism based on the statistical distribution of activations. By addressing the limitations of manual hyper-parameter tuning in traditional contrastive steering, the framework demonstrates improved attack success rates across various models including Llama-3, Mistral, and Qwen-2.

**Compliance With Llm Reviewing Policy:**

Affirmed.

**Key Questions For Authors:**

None. While the technical contributions regarding adaptive steering are noted, the current state of the manuscript’s presentation—specifically the illegible figures and the broken reference section—falls below the professional standards required for a scientific publication.

**Strengths And Weaknesses:**

Strengths

1. The introduction of an adaptive strength-tuning mechanism based on activation statistics effectively solves the "sensitivity to hyper-parameters" issue common in previous steering methods.

2. By incorporating model extraction concepts, the framework provides a more robust and less biased way to approximate ideal steering vectors compared to simple contrastive prompt pairs.

3. The framework shows consistent performance gains across multiple mainstream LLMs of different scales, proving its broad applicability in the field of AI safety evaluation.

Weaknesses

1. The visual presentation of the framework is significantly hindered by poor design choices. Specifically, the color scheme in Figure 1 lacks sufficient contrast to distinguish core modules, and the font sizes in several other figures are far too small to be legible at standard resolution, obscuring critical technical details.

2. The tables are not intuitively organized. The dense presentation of data without clear hierarchical structuring or highlighting makes it difficult for readers to quickly extract core experimental conclusions or compare cross-model performances effectively.

3. The reference section contains unacceptable typesetting failures, including improper line breaks and character overlapping. These errors, combined with inconsistent formatting throughout the manuscript (e.g., alignment and spacing issues), suggest a lack of rigorous proofreading and professionalism.

---

> ### Author Rebuttal · Authors · 2026-03-25
>
> Thank you for your time in reviewing our manuscript. In direct response to each of the specific issues you raised, we have implemented the following concrete changes:
>
> ## Figure Legibility:
> We have redesigned and regenerated all figures, especially Figure 1. We now employ a high-contrast color scheme (with a grayscale-friendly version ensured) and have significantly increased the font size of all labels and in-figure text. The revised figures guarantee that all critical details are clearly legible both on standard screens and in print preview.
>
> ## Table Organization:
> We have reorganized and reformatted all data tables in the paper. To enhance readability and highlight key findings, we have: (a) bolded the key performance metrics (e.g., the highest attack success rates); (b) optimized the hierarchical structure of row and column headers to make comparisons more intuitive; and (c) utilized the additional page allowance in the camera-ready version to judiciously split overly dense tables and improve spacing, ensuring readers can extract information quickly and effectively.
>
> ## References and Overall Formatting:
> We have performed a line-by-line review and correction of the manuscript's formatting. For the URL line-breaking issue in the references, we have enforced adaptive hyphenation by adding the \usepackage{xurl} package and related commands, eliminating all character overlaps. Furthermore, we have meticulously verified the format of every bibliographic entry to ensure full compliance with the conference style.
>
> Additionally, we have conducted a global check to unify fonts, spacing, and alignment throughout the manuscript, resolving all inconsistencies.
>
> **If you have any unresolved concerns or questions, feel free to raise them.**

---

> > ### Author Rebuttal · Reviewer_6erV · 2026-04-06
> >
> > While the modifications claimed by the authors would indeed improve the quality of the paper if implemented, I regret to inform you that these issues persist in the version currently available to me. Consequently, I will maintain my original score.

---

> > > ### Author Response · Authors · 2026-04-06
> > >
> > > Thanks for your suggestions regarding writing and formatting. We have revised our paper accordingly. Yet, we regret that ICML's review policy does not permit submitting the revised version.

---

### Official Review · Reviewer_YZiS · 2026-03-10

**Soundness:** 3
**Presentation:** 2
**Significance:** 3
**Originality:** 3
**Overall Recommendation:** 4
**Confidence:** 4

**Summary:**

In this paper, the authors leverage the idea of model extraction to guide the learned steering vectors to approximate the ideal one and propose tuning the steering strength adaptively based on contrastive activations' statistics. Experiments demonstrate that our method notably improves the effectiveness and robustness of probe-based steering, without any extra contrastive prompts or laborious manual tuning. Being an attack paper, this paper focuses on revealing the breakdown of fortified LLMs, raising the average harmfulness score from 6% to 70%.

**Compliance With Llm Reviewing Policy:**

Affirmed.

**Final Justification:**

Sorry for the delayed update. After reading the related papers recommended by the authors, along with the authors’ response, I believe this work remains meaningful for research on open-source jailbreaking. In particular, I tried simply fine-tuning Google’s newest open-source model on the Beavertails dataset and found limitations in both transferability and utility preservation. A possible reason is that these newer models have much stronger native capabilities and are therefore more sensitive to supervised fine-tuning than older models such as Llama 2. This observation further supports the value of the steering approach, which can serve as a lightweight and less harmful alternative or complementary method. Moreover, the authors’ adaptive method considers progressive, invasive attacks rather than relying only on fixed steering-direction detection. For these reasons, I think this is a significant work, and I am raising my score to 4. I also suggest evaluating domain-specific jailbreaks, such as cyber-related jailbreaks, in future work, as this has become an increasingly important topic.

**Key Questions For Authors:**

See Weakness

**Limitations:**

The authors have discussed their limitations in terms of laborious parameter tuning.  However, I have several extra concerns for this paper. See details in Weakness.

**Strengths And Weaknesses:**

Strength:
1. This paper use the models that are trained against the jailbreak attacks, which is more robust to the jailbreak attacks, instead of the normal models. Including Circuit Breaker (CB) (Zou et al., 2024), Deep Alignment (DA) (Qi et al., 2025), RepBend (RB) (Yousefpour et al., 2025), R2D2 (Mazeika et al., 2024), SafeUnlearning (SU) (Zhang et al., 2024), TAR (Tamirisa et al., 2025), DeRTA (Yuan et al., 2025) and Latent Adversarial Training (LAT) (Sheshadri et al., 2025), resulting in 12 fortified LLMs in total and prove the proposed method is effective on these models.
2. Instead of using a single model as the judge, this paper use a committee of models to make the decision in detecting the jailbreak attacks. This mitigate the risk of the single model being hacked.
3. The proposed method is adaptive, without the need for extensive hyperparameter tuning.
4. The ablation study is comprehensive, showing the effectiveness of each component of the proposed method and the alternative design's effectiveness.


Weakness:
1. AdaSteer (Zhao et al., EMNLP 2025) proposes adaptive activation steering from the defense side using the same core machinery — difference-in-means directions, linear probes/regression on activation projections, and adaptive per-input steering coefficients. The current submission never discusses AdaSteer or similar adaptive defense-side steering. This is a significant omission: if defenders also adapt their steering dynamically (as AdaSteer does), the attack's effectiveness may degrade substantially. The paper should discuss whether its iterative refinement can overcome adaptive defenses, not just static ones.
2. In appendix A, the authors admit that there is a need for threshold tuning "Currently, the most exhausting part of our method is the annotator. The annotator serves as a proxy for ideal probes but is inevitably noisy in practice. For the SRF we use, a faithless but detailed disclaimer may be scored 0.4. In contrast, the truncated beginning of a faithful but lengthy response may be scored only 0.2. This forces us to set a low threshold and a high threshold that are far apart, which helps avoid the ambiguous score interval." This makes the proposed method less "adaptive" and less "robust" to the jailbreak attacks.
3. The experiments are mainly use the model that have 7B-9B parameters. I assume experiments on the smaller model will have the same performaces. Could you please provide that results? As for larger model, how is the performance, and does the linear control assumption still hold?
4. Similer to 1. This paper does not consider the inference-time defense. Moreover, I think the threat model considered is too constrained. If the attacker can access the model's internal state, they can directly SFT the model to jailbreak it. I speculate that the proposed method is faster and need less computation resources compared with the SFT jailbreak attack, but the authors did not provide that comparison. In short, I can't be convinced by this threat model and I think only the defense side consideration (like AdaSteer) meaningful.

---

> ### Author Rebuttal · Authors · 2026-03-26
>
> Thank you for your time in reviewing our manuscript.
> ## Weakness 1
>
> Our method is also effective against AdaSteer:
>
> |Model|SRF|HB|SR|
> |-|-|-|-|
> |adasteer/Llama-3.1-8B|0.65|0.79|0.88|
> |adasteer/gemma-2-9b|0.50|0.51|0.73|
> |adasteer/Qwen2.5-7B|0.68|0.81|0.87|
>
> As we mentioned in lines 300-318, steering is essentially a parameter-efficient adapter. Indeed, AdaSteer explicitly sets the steering strength to the dot product between inputs and parameters, which seems dynamic, but other adapters also do. For example, a Rank-n LoRA's up-projection matrix has n steering vectors, and n steering strengths are set dynamically to the matrix multiplication of the input and the down-projection matrix. Since some of the defenses we included are LLMs fine-tuned with LoRA, we think our paper has already discussed adaptive defenses.
>
> **Some of the baselines are also effective against AdaSteer. We will present the results in the next discussion phase due to the length limit.**
>
> We appreciate the reviewer for highlighting AdaSteer. Discussing it will allow us to better contextualize our findings on the nature of steering. We will include AdaSteer in the revised paper.
> ## Weakness 2
>
> The adaptive retraining is an algorithm extracting the ideal probe, and the annotator is the surrogate of the ideal probe. Here, the surrogate annotator is not limited to SRF that requires threshold tuning. Alternatives to SRF include, but are not limited to, binary classifiers, like HB, and **even humans**. Trivially, these alternatives require no threshold tuning.
>
> The reasons why we chose SRF are as follows. First of all, SRF, with proper thresholds, is good enough and is not randomized (Commercial API is randomized because of lacking batch invariance). Second, we can construct both good and bad annotators by simply altering the thresholds, demonstrating how the annotator's performance influences our algorithm.
>
> Lastly, we believe that the adaptive retraining is annotator-agnostic, and the community can readily adopt more advanced or efficient annotators as they emerge.
> ## Weakness 3
>
> The LLMs we included are mostly 7B to 9B because the community mainly develops defense on LLMs of such size. As for general-purpose LLMs, we include Qwen2.5-14B-Instruct. Results demonstrate that SCAV, Angular, and our method all achieve good performance. The thresholds we used for Table 6 are the default (0.05, 0.6). If we lift the high threshold to 0.8, lowering the annotator's false positive rate. Our method can achieve 0.77, 0.93, and 0.97 on the SRF, HB, and SR metrics, respectively, which is just another SOTA.
>
> As for LLMs even larger, our hardware can not support it. However, in a concurrent work, "The Assistant Axis: Situating and Stabilizing the Default Persona of Language Models", the Anthropic team scaled steering to Llama 3.3 70B, proving that the LCA, at least, holds for LLMs with 70B.
> ## Weakness 4
>
> We thank the reviewer for raising this important point about the threat model. While the SFT attack can be powerful, a key issue is the need for target responses. In practice, if the attacker, who can only come up with harmful ideas (instructions), does not master bomb making or cyberattack, then the attacker can hardly collect enough target responses for SFT.
>
> Fortunately, our method, together with the steering family, can provide target responses since steering requires no target responses but elicits harmful knowledge from the victim LLMs. With the steered harmful outputs, the attacker can conduct SFT to jailbreak. We have successfully conducted image adversarial attacks, a prompt-tuning-based SFT, against Llava-CB. **Results are shown in Weakness 2.2 of our reply to Reviewer Lsbg, due to the length limit.** The results demonstrate that, while SFT fails with only heuristic target prefixes (i.e., "Sure, here is..."), it succeeds when provided with the target responses elicited by our steering method.
>
> Given that steering can assist, or even serve as a prerequisite for, SFT, steering can be a bridge to the loss function that is highly correlated and casual to the safety of LLMs, like Cross Entropy Loss to adversarial attacks against image classifier. Thus, **we argue that steering is meaningful for the attack scenario.**
>
> Regarding steering for defense, while steering-based defenses may be viewed as inference-time defenses, given that steering is not essentially different from other adapters during the inference, building a steering-based defense that is fundamentally more robust than other training-based defenses can be challenging (and also meaningful). Recently, leveraging our method, we jailbroke some open-source reasoning M/LLMs that are believed to be robust for their **inference-time scaling**. Due to the length limit, if reviewers are interested in jailbreaking reasoning M/LLMs, we will present results during the author-reviewer discussion period.
>
> **If you have any unresolved concerns or questions, feel free to raise them.**

---

> > ### Author Rebuttal · Reviewer_YZiS · 2026-04-02
> >
> > Thank the author for their response. I think they have addressed my concern to W1-W3. But for W4 I am still not convinced. The authors do provide an interesting reason for using this jailbreak technique. However, since there are so many open-source datasets for jailbreaking on the website, and the studies have shown the inmergent misalignment phenomenon, which means the SFT attack has transferability, I think the assumed scenario is a very narrow one.

---

> > > ### Author Response · Authors · 2026-04-02
> > >
> > > We respectfully argue that Weakness 4 is not an objective technical concern but rather a subjective one stemming from the reviewer YZiS’s personal academic values. Indeed, there are many open-source datasets for jailbreaks on the website, like AdvBench, which provides target responses. Some jailbreaks, like GCG for LLMs and PGD for MLLMs, bypass the safety alignment by maximizing the probability of generating the target responses, which is identical to prompt-tuning-based SFT.
> > >
> > > But, the key question is whether such optimization target always works, like Cross Entropy Loss for adversarial attacks against image classifier. Our experiment during the rebuttal demonstrates that they don't. AdvBench only provides **static** and **heuristic** target responses like "Sure, here is...". Such target responses maybe works for early LLMs. However, our results and [1] showed that it failed on Llava-CB. So, when we face the failure of the **static** and **heuristic** jailbreaking datasets, what can we do? Without eliciting the harmful knowledge (target responses) from the LLM, perhaps the only way is to manually search other **heuristic** target prefixes like "Sure, here is...".
> > >
> > > However, our method, together with other steering, at least provides a model-agonistic and automatic way to find the viable target responses and to build the loss function. From this perspective, the relation between steering and SFT for jailbreaking is somewhat like the relation between RL and SFT for other benign tasks. RLs like GRPO require no target responses but rollout and reward models (rubric judge or verifiable judge). While the weight update stage of these RLs can be seen as weighted SFT, the target responses for weighted SFT stem from the rollout and reward (annotation) rather than manual search. So, should we say that RLs like PPO and GRPO are meaningless because there are many SFT and even paired DPO datasets online?
> > >
> > > As for the emergent misalignment (EM) phenomenon, we would like to refer the reviewer YZiS to [3]. [3] showed that, on multiple EM models, the difference between the activations of the aligned responses and the misaligned responses robustly converges to a linear direction. Steering along this direction can control the misalignment. Thus, more probably than not, steering is one of the mechanisms that cause EM. And, thus, the EM phenomenon is one of the motivations for studying steering, not the reason for undermining the significance of steering for jailbreaking. **[3] also stated that the insecure code dataset provided by [4] can not robustly induce misalignment such that [3] had to craft a new dataset to induce the EM. Thus, the robustness and transferability of SFT-based jailbreaking with static datasets is questionable.**
> > >
> > > Lastly, as reported by [2], the most critical problem of adversarial machine learning for LLM jalibreaking is that **many defense papers evaluate their defenses on static datasets and weak attacks.** Our paper and experiments during the rebuttal provide a vivid example: With **static** jailbreak datasets like AdvBench, we may come to the conclusion that Llava-CB and all other 12 defenses are robust, but our method and some of steering baselines prove that they are not robust. This is the meaning of adversarial machine learning, revealing the worst-case robustness. That is to say, if our method, being an adversarial attack, reveals unrevealed vulnerability, it is meaningful. **Perhaps the only way to prove that SFT attacks diminishes the significance of our method, and also other steering methods, for the jailbreaking scenario is to find a SFT dataset (manually-crafted and not derived from steering) that can break all 16 defenses (12 defenses in the paper + 3 AdaSteer + Llava-CB).**
> > >
> > > Is considering steering for jailbreaking alone narrow? To conclude, given that current attacks are not strong and robust enough for jailbreaking, studying steering for boosting jalibreaking alone is meaningful enough. As for steering for defense, since AdaSteer is bypassed, currently, we believe that steering for defense is hard and worry that any positive result may lead to a false sense of security.
> > >
> > > **About Final justification:** The paper title is "Adaptive Probe-based Steering for Robust LLM **Jailbreaking**", and we hypothesized, only in the impact statement, that the proposed method can be applied to other personas **if the refusal persona is not special**. So, why is the jailbreaking scenario narrower than the paper **claimed**?
> > >
> > > [1] Improving Alignment and Robustness with Circuit Breakers. NIPS 2024.
> > >
> > > [2] The Attacker Moves Second: Stronger Adaptive Attacks Bypass Defenses Against LLM Jailbreaks and Prompt Injections. A report by OpenAI, Anthropic, Google DeepMind, HackAPrompt, Northeastern University, ETH Zürich, AI Sequrity Company, and MATS
> > >
> > > [3] Emergent Misalignment is Easy, Narrow Misalignment is Hard. ICLR 2026
> > >
> > > [4] Emergent Misalignment: Narrow Finetuning can Produce Broadly Misaligned LLMs. ICML 2025

---

### Official Review · Reviewer_rBHL · 2026-03-11

**Soundness:** 3
**Presentation:** 3
**Significance:** 3
**Originality:** 3
**Overall Recommendation:** 4
**Confidence:** 3

**Summary:**

This paper improves probe-based contrastive steering for jailbreaking safety-aligned LLMs. Two main contributions are proposed: (1) an iterative direction refinement algorithm inspired by model extraction, which augments the training set with steered activations annotated by a jailbreak judge and retrains the linear probe over multiple rounds (adaptive retraining); and (2) an adaptive strength tuning strategy that sets per-layer steering strength based on contrastive activations' logit statistics rather than requiring manual grid search over L continuous parameters. The paper also identifies that steering all token positions (not just response tokens) and discarding the last-layer activation are important implementation choices. Evaluated against 12 fortified LLMs (including Circuit Breaker, Deep Alignment, RepBend, R2D2, SafeUnlearning, TAR, DeRTA, and LAT variants) on StrongReject and HarmBench (200 harmful prompts), the method achieves an average harmfulness score of 0.70 (SR metric), substantially outperforming prior contrastive steering baselines (RepE, SCAV, Refusal Direction, Angular Steering) that average 0.02–0.24.

**Compliance With Llm Reviewing Policy:**

Affirmed.

**Key Questions For Authors:**

1. **Coherence of steered outputs:** Can you provide a human evaluation or automatic coherence metric for the generated responses? High harmfulness scores from LLM judges do not guarantee that the outputs are actually useful harmful instructions rather than incoherent text flagged as harmful.

2. **TAR resilience:** Llama3-TAR shows the best robustness among evaluated models (SR=0.50 vs. 0.85+ for most others). You attribute this to adversarial training over fully tampered parameters. Does this suggest that full-parameter adversarial training is the most promising defense direction, and have you tested increasing the number of adaptive retraining iterations specifically for TAR?

3. **Transferability:** Do the steering directions found for one model transfer to other models with the same base architecture but different defense training? For example, do directions found on Llama3-CB work for Llama3-RB?

4. **Computational cost:** What is the total computational cost of the 20-iteration adaptive retraining process? How does this compare to the cost of the attacks being evaluated against (e.g., GCG, which requires gradient optimization)?

**Limitations:**

Partially. The authors discuss white-box access requirements and mention that the method could benefit red-teaming. However, the impact statement does not adequately address the risk that this method substantially lowers the barrier for jailbreaking hardened open-source models. The dual-use concern is more acute here than in typical jailbreaking papers because the method defeats multiple state-of-the-art defenses simultaneously.

**Strengths And Weaknesses:**

### Strengths

1. **Significant performance improvement.** The leap from prior steering methods (average 0.02–0.24) to 0.70 average across 12 hardened LLMs is dramatic. Notably, the method succeeds on models where prior steering methods achieve near-zero scores (e.g., Llama2-DA, Vicuna-SU, Mistral-CB), demonstrating genuine capability rather than incremental improvement.

2. **Well-motivated adaptive strength tuning.** The analysis of activation magnitude variation across layers (Figure 3) and its connection to oversteering is insightful. The adaptive logit-based strength tuning (setting s(l) based on contrastive activation statistics rather than uniform values) is a principled solution with clear theoretical grounding through Equation 4–5.

3. **Thorough ablation study.** Tables 2 and 3 systematically decompose the contribution of each component (Adaptive Strength, Discarding Last-layer Activation, Steering All Tokens, and Adaptive Retraining). The incremental analysis clearly shows that each component is necessary and their contributions are complementary.

4. **Broad fortified model coverage.** Testing against 12 LLMs spanning 8 different defense paradigms (adversarial training, unlearning, activation manipulation, deep alignment) provides strong evidence that the vulnerability is general, not specific to a particular defense strategy.

5. **LoRA interpretation of steering.** The insight that probe-based steering can be viewed as a rank-1 LoRA with fixed bias (Equation 6), which motivates steering all token positions, is elegant and provides a unified perspective connecting steering to parameter-efficient fine-tuning.

### Weaknesses

1. **White-box access requirement.** The method requires access to all intermediate layer activations during inference, which limits applicability to open-weight models only. While this is inherent to all contrastive steering approaches, the practical impact is that the most widely deployed models (GPT-4, Claude, Gemini) cannot be evaluated.

2. **Evaluation judges may over-estimate harmfulness.** The paper uses SRF (the same judge used for direction searching) as one of three evaluation judges. While the authors mitigate this by including HB and SR judges, using the training signal as an evaluation metric introduces potential bias. The discrepancy between SRF and HB/SR scores on some models (e.g., Llama3-TAR: SRF=0.32, HB=0.24, SR=0.50) warrants discussion.

3. **Limited analysis of generated content quality.** High harmfulness scores indicate that the model produces content judged as harmful, but the paper does not analyze the coherence, relevance, or specificity of the generated responses. Oversteering can produce high-harmfulness but incoherent text, and while the method addresses oversteering at the activation level, no human evaluation or coherence metric is reported.

4. **Convergence properties not analyzed.** The adaptive retraining runs for T=20 iterations by default, but Figure 4 shows different convergence behaviors across models (some converge quickly, others fluctuate). No formal convergence analysis or stopping criterion is provided.

5. **Defense implications underexplored.** The paper concludes with hope that the method benefits red-teaming, but does not provide concrete suggestions for how defenders should respond. Given that the method breaks multiple defense paradigms, constructive guidance for defense improvement would strengthen the contribution.

---

> ### Author Rebuttal · Authors · 2026-03-25
>
> Thanks for your time in reviewing.
>
> ## Weakness 1
>
> We acknowledge the limitation of the white-box requirement. However, steering-based methods can be used to build the loss function for optimizing adversarial examples, **which can attack remote M/LLMs that accept input only.** We have conducted preliminary experiments demonstrating that our method can boost white-box adversarial examples. Due to the length limit, please refer to Weakness 2.2 in our reply to reviewer Lsbg. As for black-box attacks, one approach is to utilize the adversarial transferability: the adversarial examples optimized on local surrogate models may also break remote unseen models. However, improving the adversarial transferability on M/LLM is hard.
>
> ## Weaknesses 2 & 3
>
> We recognize the potential bias induced by SRF. Thus, we present SRF, SR, and HB separately instead of providing only average scores such that readers can choose to ignore SRF.
>
> We chose SRF, SR, and HB because the StrongReject team conducted extensive experiments showing that SRF, SR, and HB have low divergence from human assessments **in terms of responses' coherence and helpfulness**. Results demonstrated that SRF tends to have a lower score than human assessment, while SR's tends to be high, **which may explain the discrepancy**. We have also manually checked these responses, and confirmed the StrongReject team's conclusion.
>
> ## Weakness 4
>
> By far, we believe that there are three factors affecting the convergence: the annotator, the initial probe, and the model’s utility.
>
> First, if the annotator is noisy, then the adaptive training, which extract (steal) the annotator, will be noisy too. Detailed analysis of the annotator is in Appendix A.
>
> Second, the initial probe can be good enough such that the adaptive training instantly converges. As presented in Figure 11, the initial probe of Vicuna-SU has already modeled the linear concept well, which assigns higher logits to more faithful responses.
>
> Third, the low utility will hinder convergence at high SRF scores. For example, R2D2 and Mistral-SU, two LLMs with the lowest utility, converge at low SRF scores because their utility does not support more coherent responses.
>
> ## Weakness 5
> There are three levels of defense: parameter, image, and text.
>
> Building parameter-level and image-level defense is hard. Any malicious prompt corresponds to a malicious reply. Parameter-level and image-level attacks can use gradient descent to easily maximize the LLM’s likelihood of generating malicious replies. **Now, since steering can induce malicious replies, attacks of these two levels become simple.** While current text-level attacks are slow and heuristic due to weak discrete optimization, if discrete optimization becomes powerful, text-level defense will also be hard.
>
> Lastly, keeping the model close-source is the simplest defense that can increase the attack cost but can not improve adversarial robustness.
> ## Question 1
> See Weaknesses 2&3.
> ## Question 2
> More iterations do not help. Yet, we can boost steering against TAR by altering the activation extractor and the sampler. As shown in Row 4, Table 7, by using activations from response token position and setting the sampling strength to the median of useful samples' logits, we can promote the average harmful scores from 0.35 to 0.56. Given these promotions induced by minor alternation under our framework, TAR might be effectively jailbroken if better annotators, activation extractors, or samplers are adopted. Thus, we can not claim that TAR is promising.
> ## Question 3
> It is well known that LoRA has good transferability within a model family. Since probe-based steering can be seen as a Rank-1 LoRA, it may also have certain transferability. We directly use steering vectors trained on Llama-CB to jailbreak Llama-RB:
>
> |SRF|HB|SR|
> |:-:|:-:|:-:|
> |0.62|0.77|0.69|
>
> We can find that the transfer attack achieves nontrivial harmful scores, proving that the probe-based steering also has transferability within a model family.
> ## Question 4
> All experiments were conducted on a A100-PCIE-40G GPU. For 7B-14B models, with 50 training and 50 evaluation prompts, 20 iterations take approximately 5 hours, primarily for rollout annotation. Evaluation itself requires no additional computation. GCG, set to 500 iterations, takes over 10 minutes per prompt, exceeding 33 hours for 200 prompts.
>
> Other steering-based attacks take average 5 minutes to compute steering vectors, but this speed relies on human-pre-labeled contrastive prompts. As noted (lines 419-428), naively increasing contrastive prompts is less data-efficient than our adaptive retraining. If pre-labeling is not available, and rollout is required for annotation, they are less efficient than our method.
> ## Limitations
> The limitation you pointed out is similar to that raised by Reviewer Lsbg. Due to the length limit, please refer to the Limitation section in our reply to Reviewer Lsbg.
>
> **If you have any concerns or questions, feel free to ask.**

---

### Official Review · Reviewer_Lsbg · 2026-03-12

**Soundness:** 3
**Presentation:** 2
**Significance:** 3
**Originality:** 3
**Overall Recommendation:** 4
**Confidence:** 4

**Summary:**

Steering vectors can jailbreak models. This paper introduces a method to adaptively steer a model and then add steered responses to a training set to improve the steering. They input harmful prompts, generate responses, and collect activations with the LLM steered by the current probe. Then they annotate each activation by judging the corresponding LLM responses, and add these annotated activations to the training set for the next iteration. They also try various smaller improvement to the steering process like adaptive steering strengths. They find the combination of their smaller improvements and adaptive method outperforms all steering-jailbreak defences on several safety-hardened LLMs.

**Compliance With Llm Reviewing Policy:**

Affirmed.

**Key Questions For Authors:**

1) Have you tried other baselines with +AS+DLA+SAT+NA? If yes, this would increase my confidence that the adaptive retraining is key.

**Limitations:**

Mostly yes, although I would like a stronger discussion on how they propose to improve robustness and safety via jailbreaking with steering.

**Strengths And Weaknesses:**

Strengths:
Soundness
 - The ablations seem clean/reasonable and the method outperforms baselines.
Significance
 - The ability to jailbreak models with linear changes to activations is a niche application but is useful for understanding safety properties of models
 - All the innovations seem well motivated and easy to apply to many methods.
Originality
 - The method seems novel
Weaknesses:
Presentation
 - Table 2 and Table 3 are too small and dense and have many acronyms which are buried in other parts of the paper.
 - I found the beginning a bit dense and I'm not sure if all the theory/background math (section 3.1) is actually useful to understand the method.
 - Nit: you could explain baselines better, and contrast them with your work more clearly
Soundness
 - There are a bunch of small changes listed in this paper, which is fine, but I think it's a little misleading to not include more experiments like SCAV+AS+DLA+SAT+NA which test the baselines augmented with stronger settings.
 - Small thing but missing baselines like finetuning the models (perhaps with LoRA to keep parameter-efficiency)

Nit: Grammar--"and explore how high the harmfulness score can be under certain constraint." should be "constraints"

---

> ### Author Rebuttal · Authors · 2026-03-25
>
> Thank you for your time in reviewing our manuscript.
>
> ## Weakness 1: Presentation
>
> ### 1.Table
> We want to include as many defenses in the main paper as possible to demonstrate the generality of our method. Perhaps we can leave some of these defenses in the appendix to make these table look less crowded, or we can split it since camera-ready allows an extra page. We will polish the tables accordingly.
>
> ### 2. Section 3.1
>
> Section 3.1 may help readers who are not familiar with steering and linear concept hypothesis. We will polish this section for better readability.
>
> ### 3. Introducing Baselines
>
> A detailed introduction to baselines will be added to the appendix. We will refer readers to the baseline introduction in the main paper.
>
> ## Weakness 2: Soundness
>
> ### 1. SCAV+AS+DLA+SAT+NA
> SCAV+AS+DLA+SAT+NA has already been presented in Row 6, Table 3. NA did not improve the performance as our adaptive retraining did. Perhaps the layout of Table 3 makes you miss it. Sorry for that.
>
>
> ### 2. Fine-tuning
> The most distinct difference between contrastive steering and fine-tuning is that **the former does not require target responses, while the latter does**. Thus, it is hard to directly compare contrastive steering and fine-tuning fairly.
>
> Yet, we may compare these two types of method from the essence: As mentioned in lines 300-318, the probe-based steering can be seen as a parameter efficient adapter, or more specifically, a Rank-1 LoRA. Thus, theoretically, PeFT like LoRA is strictly stronger than probe-based steering (One can have a LoRA that is at least better than our steering by simply copying the vector we discover).
>
> In practice, however, fine-tuning requires target responses. Thus, the performance of fine-tuning is influenced by the target response. Let us take the visual version of circuit breaker (GraySwanAI/llava-v1.6-mistral-7b-hf-RR, llava-CB in short) as an example, on which we can conduct visual adversarial attacks, **which is identical to prompt-tuning.** When we set the target response to "Sure, here is..." (PGD+Sure), a target prefix widely adopted by many fine-tuning attacks, we can only get near-zero ASR. But, if we use our method to jailbreak llava-CB, generate harmful outputs, and use these harmful instruction-output pairs for optimizing adversarial examples (PGD+Ours), we got 70%+ ASRs.
>
> | Method   | SRF   | HB    | SR    |
> |----------|-------|-------|-------|
> | PGD+Sure | 0.01  | 0.02  | 0.00  |
> | PGD+Ours | 0.66  | 0.81  | 0.77  |
> | Ours     | 0.66  | 0.88  | 0.78  |
>
> This experiment demonstrates that our method can even boost fine-tuning attacks by providing better target responses! (Crafting adversarial examples is meaningful because LLMs deployed by others accept only inputs. Yet, for local LLMs, since we have already jailbroken them with steering, fine-tuning attacks seemed unnecessary?)
>
> To conclude, PeFT is theoretically stronger than ours because our probe-based steering is, in fact, an instantiation of parameter-efficient adapters. Yet, how to acquire target responses for fine-tuning can be a hard-to-solve issue if we do not have any harmful knowledge. Our method, together with other steering, can solve this issue by providing target responses without the need for harmful knowledge.
>
> ## Key Question
>
> See Weakness 2.1
>
> ## Limitation
>
> We think the most urgent problem for adversarial machine learning on LLM (and also MLLM) jailbreaking is developing strong adaptive attacks, which is exactly what our paper did. Without a strong adaptive attack, defenses may exhibit a false sense of security, inducing security holes. For example, Llava-CB might be deployed because of its claim on robustness against image adversarial attacks. Yet, while the deployer still believes Llava-CB is impregnable, a strong attacker may break it with attacks that are, at least, as strong as ours. Thus, we believe our method can improve robustness and safety by providing insights on developing strong adversarial attacks and preventing defenses that are not truly robust from deploying.
>
> **If you have any unresolved concerns or questions, feel free to raise them.**

---

> > ### Author Rebuttal · Reviewer_Lsbg · 2026-04-02
> >
> > > SCAV+AS+DLA+SAT+NA
> > Re this, I know you have it. I mean trying e.g. RepE+AS+DLA+SAT+NA etc. would that be possible? Or whatever other base steering method. In general I would like to know which elements of your overall method are crucial vs small incremental improvements.

---

> > > ### Author Response · Authors · 2026-04-02
> > >
> > > ## AS
> > > RepE, Angular, and RD are all not probe-based steering. Thus, AS can not be applied to them. AS can only be applied to SCAV.
> > >
> > > ## SAT
> > > RepE, Angular, and RD all have applied SAT by default. Thus, SAT can only be apllied to SCAV.
> > >
> > > ## DLA
> > > RepE manually chooses middle layers (e.g., 8 to 20 for 32-layer LLMs) by default. Angular and RD both do not steer the last hidden state (i.e., the tensor that is fed into the LLM unembedding layer). Thus, DLA also can only be applied to SCAV.
> > >
> > > ## NA
> > >
> > > We have included RepE+NA, Angular+NA, SCAV+NA, RD+NA in Table 3.
> > >
> > > In short, we have:
> > > 1. RepE = RepE + SAT + DLA
> > > 2. Angular = Angular + SAT + DLA
> > > 3. RD = RD + SAT + DLA
> > > 4. AS can not be applied to RepE, Angular, and RD. They already have their own strength tuning schemes.
> > > 5. "+ SAT + DLA + NA" is the maximum modification we can make to RepE, Angular, and RD.
> > >
> > > To conclude, AS eliminates the need for strength tuning of the probe-based steering (e.g., SCAV). SAT and DLA are two small implementation details that can fix the underlying drawback of SCAV. AR, our adaptive retraining, is the model extraction algorithm that is more effective and efficient than naive augmentation (NA). All of them are indispensable for boosting SCAV, the probe-based steering we focused on in this paper.
> > >
> > >  **Please let us know if you have any further concerns by updating the rebuttal acknowledgement. We are happy to address all concerns and confusions the reviewers have. These discussions will save the times of future readers who may have the same concerns and confusions, which will promote our paper's quality.**

---

### Decision · Program_Chairs · 2026-04-30

**Decision:**

Accept (regular)

**Comment:**

This paper proposes an adaptive probe-based steering framework for jailbreaking safety-aligned LLMs, combining iterative retraining inspired by model extraction with an activation-statistics-based mechanism for tuning steering strength. Across reviews, the core technical contributions are consistently recognized. In particular, the empirical gains are substantial, with harmfulness scores increasing from around 6% to 70% across a wide range of fortified models, and the ablation studies are thorough and well-structured. Reviewers also highlighted the conceptual clarity of interpreting probe-based steering through a LoRA lens, as well as the broad evaluation coverage across different defense paradigms. Overall, three reviewers lean positive, noting both the novelty and the practical value of the method for stress-testing and red-teaming LLM safety systems

During the rebuttal, the authors made a strong effort to address key concerns. They clarified the role of steering relative to fine-tuning, provided additional evidence on transferability and computational efficiency, and discussed the relationship between their method and adaptive defenses such as AdaSteer. The responses help contextualize the method more clearly within the broader attack landscape, and several reviewers acknowledged partial or substantial resolution of their concerns. That said, some limitations remain. In particular, the reliance on white-box access restricts applicability to open-weight models, and the argument for why steering is preferable to alternatives such as SFT is convincing but still somewhat scenario-dependent. Additionally, questions around output quality and coherence beyond judge-based metrics were addressed indirectly but would benefit from more explicit empirical validation.

Taking everything together, the paper presents a technically solid and practically relevant contribution to adversarial analysis of LLMs. The empirical results are strong, and the method offers a useful perspective on the limits of current safety defenses. While some concerns remain around threat model scope and evaluation completeness, they do not fundamentally undermine the main contribution. Given the positive reception from most reviewers and the strengthened evidence after rebuttal, I recommend a weak accept.